# UniPC: A Unified Predictor-Corrector Framework for Fast Sampling of Diffusion Models

**Wenliang Zhao**[*] **Lujia Bai**[*] **Yongming Rao** **Jie Zhou** **Jiwen Lu**[†]

Tsinghua University

## Abstract

Diffusion probabilistic models (DPMs) have demonstrated a very promising ability in high-resolution image synthesis. However, sampling from a pre-trained DPM is time-consuming due to the multiple evaluations of the denoising network, making it more and more important to accelerate the sampling of DPMs. Despite recent progress in designing fast samplers, existing methods still cannot generate satisfying images in many applications where fewer steps (*e.g.*, <10) are favored. In this paper, we develop a unified corrector (UniC) that can be applied after any existing DPM sampler to increase the order of accuracy without extra model evaluations, and derive a unified predictor (UniP) that supports arbitrary order as a byproduct. Combining UniP and UniC, we propose a unified predictor-corrector framework called UniPC for the fast sampling of DPMs, which has a unified analytical form for any order and can significantly improve the sampling quality over previous methods, especially in extremely few steps. We evaluate our methods through extensive experiments including both unconditional and conditional sampling using pixel-space and latent-space DPMs. Our UniPC can achieve 3.87 FID on CIFAR10 (unconditional) and 7.51 FID on ImageNet 256×256 (conditional) with only 10 function evaluations. Code is available at `https://github.com/wl-zhao/UniPC`.

## 1 Introduction

Diffusion probabilistic models (DPMs) [33, 13, 35] have become the new prevailing generative models and have achieved competitive performance on many tasks including image synthesis [8, 29, 13], video synthesis [15], text-to-image generation [27, 29, 12], voice synthesis [5], *etc*. Different from GANs [10] and VAEs [20], DPMs are trained to explicitly match the gradient of the data density (*i.e.*, score), which is more stable and less sensitive to hyper-parameters. However, sampling from a pre-trained DPM usually requires multiple model evaluations to gradually perform denoising from Gaussian noise [13], consuming more inference time and computational costs compared with single-step generative models like GANs.

Recently, there have been increasing efforts to accelerate the sampling of DPMs [31, 28, 34, 25, 40]. Among those, training-free methods [34, 25, 40] enjoy a wider usage in applications because they can be directly applied to off-the-shelf pre-trained DPMs. Although these methods have significantly reduced the sampling steps from 1000 to less than 20 steps, the sampling quality with extremely few steps (*e.g.*, <10) has been rarely investigated. Few-step sampling can be used in many scenarios where we need to efficiently obtain plausible samples, such as designing a proper prompt for a text-to-image diffusion model [29, 30] and computing losses on the generated samples during the training of some diffusion-based visual systems [1, 6]. However, current fast samplers usually struggle to generate high-quality samples within 10 steps (see Figure 1).

---

[*]Equal contribution. [†]Corresponding author.

37th Conference on Neural Information Processing Systems (NeurIPS 2023).

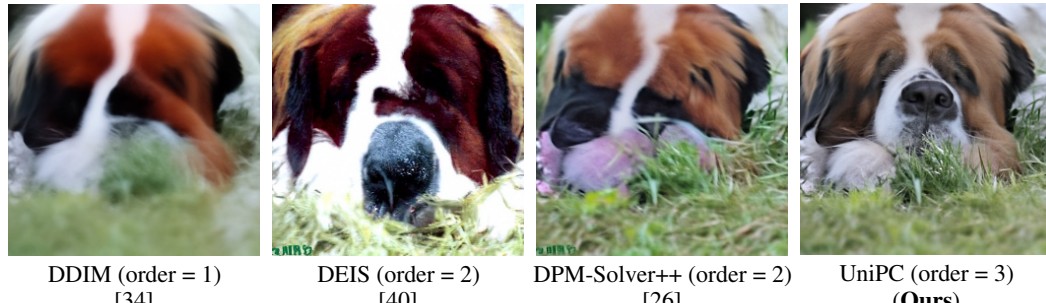

| DDIM (order = 1) [34] | DEIS (order = 2) [40] | DPM-Solver++ (order = 2) [26] | UniPC (order = 3) (**Ours**) |

Figure 1: **Qualitative comparisons between our UniPC and previous methods.** All images are generated by sampling from a DPM trained on ImageNet $256 \times 256$ with only **7** number of function evaluations (NFE) and a classifier scale of 8.0. We show that our proposed UniPC can generate more plausible samples with more visual details compared with previous first-order sampler [34] and high-order samplers [40, 26]. Best viewed in color.

In this paper, we propose a training-free framework for the fast sampling of DPMs called ***UniPC***. We find that UniPC significantly outperforms existing methods within 5~10 NFE (number of function evaluations), and can also achieve better sampling quality with more sampling steps. Specifically, we first develop a unified corrector (UniC) which works by using the the model output $\boldsymbol{\epsilon}_\theta(\boldsymbol{x}_{t_i}, t_i)$ at the current timestep $t_i$ to obtain a refined $\boldsymbol{x}_{t_i}^c$. Different from the predictor-corrector paradigm in numerical ODE solver that requires a doubled NFE, our UniC reuses the model output $\boldsymbol{\epsilon}_\theta(\boldsymbol{x}_{t_i}, t_i)$ to the next sampling step, thus introducing no extra function evaluation. UniC can be applied after any existing DPM sampler to increase the order of accuracy, while the inference speed is almost unaffected. Interestingly, we also find that by simply changing a hyper-parameter in UniC, a new family of predictors (UniP) can be further obtained.

Since our UniC is method-agnostic, we combine UniP and UniC to obtain a new family of fast samplers called UniPC. Different from previous fast solvers [26, 25, 40] that either have no higher-order (*e.g.*, $> 3$) variants or have no explicit forms, our UniPC supports arbitrary orders with a unified analytical expression and are easy to implement. Benefiting from the universal design, variants of UniPC (*e.g.*, singlestep/multistep, noise/data prediction) can be easily derived. We theoretically prove that UniPC enjoys higher convergence order and empirically demonstrate that UniPC has better sampling quality in a variety of scenarios. We also show that the inference speed and memory usage of UniPC is similar to DPM-Solver++ [26], indicating that UniPC can achieve superior performance under the same computational budgets.

We conduct extensive experiments with both pixel-space and latent-space DPMs to verify the effectiveness of the proposed UniPC. Our results show that UniPC performs consistently better than previous state-of-the-art methods on both unconditional and conditional sampling tasks. Notably, UniPC can achieve 3.87 FID on CIFAR10 (unconditional) and 7.51 FID on ImageNet $256 \times 256$ (conditional) with only 10 function evaluations. We also demonstrate that UniC can improve the sampling quality of several existing fast samplers significantly with very few NFE (number of function evaluations). Some qualitative comparisons are shown in Figure 1, where we observe that our UniPC can generate images with more visual details than other methods.

## 2 Background and Related Work

### 2.1 Diffusion Probabilistic Models

For a random variable $\boldsymbol{x}_0$ with an unknown distribution $q_0(\boldsymbol{x}_0)$, Diffusion Probabilistic Models (DPMs) [33, 13, 19] transit $q_0(\boldsymbol{x}_0)$ at time 0 to a normal distribution $q_T(\boldsymbol{x}_T) \approx \mathcal{N}(\boldsymbol{x}_T | \boldsymbol{0}, \tilde{\sigma}^2 \boldsymbol{I})$ at time $T$ for some $\tilde{\sigma} > 0$ by gradually adding Gaussian noise to the observation $\boldsymbol{x}_0$. For each time $t \in [0, T]$, and given $\sigma_t, \alpha_t > 0$, the Gaussian transition is

$$q_{t|0}(\boldsymbol{x}_t | \boldsymbol{x}_0) = \mathcal{N}(\boldsymbol{x}_t | \alpha_t \boldsymbol{x}_0, \sigma_t^2 \boldsymbol{I}),$$

where $\alpha_t^2 / \sigma_t^2$ (the *signal-to-noise-ratio* (SNR)) is strictly decreasing w.r.t. $t$ [19].

Let $\epsilon_\theta(\boldsymbol{x}_t, t)$ denote the noise prediction model using data $\boldsymbol{x}_t$ to predict the noise $\epsilon$, and the parameter $\theta$ is obtained by minimizing

$$\mathbb{E}_{\boldsymbol{x}_0, \epsilon, t}[\omega(t)\|\epsilon_\theta(\boldsymbol{x}_t, t) - \epsilon\|_2^2],$$

where $\boldsymbol{x}_0 \sim q_0(\boldsymbol{x}_0)$, $t \in \mathcal{U}[0, T]$, and the weight function $\omega(t) > 0$. Sampling from DPMs can be achieved by solving the following diffusion ODEs [35]:

$$\frac{\mathrm{d}\boldsymbol{x}_t}{\mathrm{d}t} = f(t)\boldsymbol{x}_t + \frac{g^2(t)}{2\sigma_t}\epsilon_\theta(\boldsymbol{x}_t, t), t \in [0, T], \quad \boldsymbol{x}_T \sim \mathcal{N}(\boldsymbol{0}, \tilde{\sigma}^2\boldsymbol{I}) \tag{1}$$

where $f(t) = \frac{\mathrm{d}\log\alpha_t}{\mathrm{d}t}$, $g^2(t) = \frac{\mathrm{d}\sigma_t^2}{\mathrm{d}t} - 2\frac{\mathrm{d}\log\alpha_t}{\mathrm{d}t}\sigma_t^2$.

## 2.2 Fast Sampling of DPMs

Fast samplers of DPMs can be either training-based [31, 2, 37] or training-free [25, 26, 40, 24, 41]. Training-based samplers require further training costs while training-free methods directly use the original information without re-training and are easy to implement in conditional sampling. The essence of training-free samplers is solving stochastic differential equations (SDEs)[13, 35, 3, 41] or ODEs[26, 40, 24, 34, 25]. Other fast sampling methods include modifying DPMs [9] and the combination with GANs [38, 36].

Among others, samplers solving diffusion ODEs are found to converge faster for the purpose of sampling DPMs [34, 35]. Recent works [40, 25, 26] show that ODE solvers built on exponential integrators [17] appear to have faster convergence than directly solving the diffusion ODE (1). The solution $\boldsymbol{x}_t$ of the diffusion ODE given the initial value $\boldsymbol{x}_s$ can be analytically computed as [25]:

$$\boldsymbol{x}_t = \frac{\alpha_t}{\alpha_s}\boldsymbol{x}_s - \alpha_t \int_{\lambda_s}^{\lambda_t} e^{-\lambda}\hat{\epsilon}_\theta(\hat{\boldsymbol{x}}_\lambda, \lambda)\mathrm{d}\lambda, \tag{2}$$

where we use the notation $\hat{\epsilon}_\theta$ and $\hat{\boldsymbol{x}}_\lambda$ to denote changing from the domain of time($t$) to the domain of half log-SNR($\lambda$), i.e., $\lambda_t = \log(\alpha_t/\sigma_t)$, $\hat{\boldsymbol{x}}_\lambda := \boldsymbol{x}_{t_\lambda(\lambda)}$ and $\hat{\epsilon}_\theta(\cdot, \lambda) := \epsilon_\theta(\cdot, t_\lambda(\lambda))$.

Based on the exponential integrator, [25] proposes to approximate $\hat{\epsilon}_\theta$ via taylor expansion and views DDIM as DPM-Solver-1, i.e.,

$$\tilde{\boldsymbol{x}}_{t_i} = \frac{\alpha_{t_i}}{\alpha_{t_{i-1}}}\tilde{\boldsymbol{x}}_{t_{i-1}} - \sigma_{t_i}(e^{\lambda_{t_i} - \lambda_{t_{i-1}}} - 1)\epsilon_\theta(\hat{\boldsymbol{x}}_{t_{i-1}}, t_{i-1}). \tag{3}$$

[26] considers rewriting (2) using $\hat{\boldsymbol{x}}_\theta$ instead of $\hat{\epsilon}_\theta$; [40] derives the taylor expansion formulae with respect to $t$ instead of the half log-SNR($\lambda$). [24] employs pseudo numerical methods such as Runge-Kutta method directly for the updating of $\epsilon_\theta$ of (3). Although many aforementioned high-order solvers are proposed, existing solvers of diffusion ODEs can be explicitly computed for orders not greater than 3, due to the lack of analytical forms.

# 3 A Unified Predictor-Corrector Solver

In this section, we propose a unified predictor-corrector solver of DPMs called UniPC, consisting of UniP and UniC. Our UniPC is unified in mainly two aspects: 1) the predictor (UniP) and the corrector (UniC) share the same analytical form; 2) UniP supports arbitrary order and UniC can be applied after off-the-shelf fast samplers of DPMs to increase the order of accuracy.

## 3.1 The Unified Corrector UniC-$p$

Modern fast samplers based on discretizing diffusion ODEs [25, 34, 40] aim to leverage the previous $p$ points $\{\tilde{\boldsymbol{x}}_{t_{i-k}}\}_{k=1}^p$ to estimate $\tilde{\boldsymbol{x}}_{t_i}$ with $p$ order of accuracy. Despite the rapid development of fast samplers, the quality of few-step sampling still has room for improvement. In this paper, we propose a corrector called UniC-$p$ to improve the initial estimation using not only the previous $p$ points but also the current point. Formally, after obtaining the initial estimation $\tilde{\boldsymbol{x}}_{t_i}$, we perform the correction step through the following formula:

$$\tilde{\boldsymbol{x}}_{t_i}^c = \frac{\alpha_{t_i}}{\alpha_{t_{i-1}}}\tilde{\boldsymbol{x}}_{t_{i-1}}^c - \sigma_{t_i}(e^{h_i} - 1)\epsilon_\theta(\tilde{\boldsymbol{x}}_{t_{i-1}}, t_{i-1}) - \sigma_{t_i}B(h_i)\sum_{m=1}^p \frac{a_m}{r_m}D_m, \tag{4}$$

**Algorithm 1** UniC-$p$

**Require:** $\{r_i\}_{i=1}^{p-1}$, $\epsilon_\theta$ network, any $p$-order solver Solver-p, a buffer $Q = \{\epsilon_\theta(\tilde{\boldsymbol{x}}_{t_{i-k}}, t_{i-k})\}_{k=1}^p$.
$h_i \leftarrow \lambda_{t_i} - \lambda_{t_{i-1}}$, $\tilde{\boldsymbol{x}}_{t_i} \leftarrow$ Solver-p$(\tilde{\boldsymbol{x}}_{t_{i-1}}^c, Q)$
**for** $m = 1$ **to** $p$ **do**
   $s_m \leftarrow t_\lambda(r_m h + \lambda_{t_{i-1}})$
   $D_m \leftarrow \epsilon_\theta(\tilde{\boldsymbol{x}}_{s_m}, s_m) - \epsilon_\theta(\tilde{\boldsymbol{x}}_{t_{i-1}}, t_{i-1})$
**end for**
Compute $\boldsymbol{a}_p \leftarrow \boldsymbol{R}_p^{-1}(h_i)\boldsymbol{\phi}_p(h_i)/B(h_i)$, where $\boldsymbol{R}_p, \boldsymbol{\phi}_p$ are as defined in Theorem 3.1
$\tilde{\boldsymbol{x}}_{t_i}^c \leftarrow \frac{\alpha_{t_i}}{\alpha_{t_{i-1}}}\tilde{\boldsymbol{x}}_{t_{i-1}}^c - \sigma_{t_i}(e^{h_i} - 1)\epsilon_\theta(\tilde{\boldsymbol{x}}_{t_{i-1}}, t_{i-1})$
      $- \sigma_{t_i}B(h_i)\sum_{m=1}^p a_m D_m/r_m$
$Q \overset{\text{buffer}}{\leftarrow} \epsilon_\theta(\tilde{\boldsymbol{x}}_{t_i}, t_i)$
**return:** $\tilde{\boldsymbol{x}}_{t_i}^c$

**Algorithm 2** UniP-$p$

**Require:** $\{r_i\}_{i=1}^{p-1}$, $\epsilon_\theta$ network, a buffer $Q = \{\epsilon_\theta(\tilde{\boldsymbol{x}}_{t_{i-k}}, t_{i-k})\}_{k=1}^p$.
$h_i \leftarrow \lambda_{t_i} - \lambda_{t_{i-1}}$
**for** $m = 1$ **to** $p - 1$ **do**
   $s_m \leftarrow t_\lambda(r_m h + \lambda_{t_{i-1}})$
   $D_m \leftarrow \epsilon_\theta(\tilde{\boldsymbol{x}}_{s_m}, s_m) - \epsilon_\theta(\tilde{\boldsymbol{x}}_{t_{i-1}}, t_{i-1})$
**end for**
Compute $\boldsymbol{a}_{p-1} \leftarrow \boldsymbol{R}_{p-1}^{-1}(h_i)\boldsymbol{\phi}_{p-1}(h_i)/B(h_i)$, where $\boldsymbol{R}_{p-1}, \boldsymbol{\phi}_{p-1}$ are as defined in Theorem 3.1
$\tilde{\boldsymbol{x}}_{t_i} \leftarrow \frac{\alpha_{t_i}}{\alpha_{t_{i-1}}}\tilde{\boldsymbol{x}}_{t_{i-1}} - \sigma_{t_i}(e^{h_i} - 1)\epsilon_\theta(\tilde{\boldsymbol{x}}_{t_{i-1}}, t_{i-1})$
      $- \sigma_{t_i}B(h_i)\sum_{m=1}^{p-1} a_m D_m/r_m$
$Q \overset{\text{buffer}}{\leftarrow} \epsilon_\theta(\tilde{\boldsymbol{x}}_{t_i}, t_i)$
**return:** $\tilde{\boldsymbol{x}}_{t_i}$

where $\tilde{\boldsymbol{x}}_{t_i}^c$ denotes the corrected result, $B(h) = \mathcal{O}(h)$ is a non-zero function of $h$, $h_i = \lambda_{t_i} - \lambda_{t_{i-1}}$ is the step size in the half-log-SNR($\lambda$) domain, $r_1 < r_2 < \cdots < r_p = 1$ are a non-zero increasing sequence, determining which previous points are used. Specifically, we use $\{r_i\}_{m=1}^p$ to interpolate between $\lambda_{t_{i-1}}$ to $\lambda_{t_i}$ to obtain the auxiliary timesteps $s_m = t_\lambda(r_m h + \lambda_{t_{i-1}}), m = 1, 2, \ldots, p$. The model outputs at these timesteps are used to compute $D_m$ by

$$D_m = \epsilon_\theta(\tilde{\boldsymbol{x}}_{s_m}, s_m) - \epsilon_\theta(\tilde{\boldsymbol{x}}_{t_{i-1}}, t_{i-1}). \tag{5}$$

We now describe how to choose $\{a_m\}_{m=1}^p$ in UniC-$p$ to effectively increase the order of accuracy. The main idea is to cancel out low-order terms between the numerical estimation (4) and the theoretical solution (2). In practice, we expand the exponential integrator in (2) as follows:

$$\boldsymbol{x}_{t_i} = \frac{\alpha_{t_i}}{\alpha_{t_{i-1}}}\boldsymbol{x}_{t_{i-1}} - \sigma_{t_i}(e^{h_i} - 1)\epsilon_\theta(\boldsymbol{x}_{t_{i-1}}, t_{i-1})$$
$$- \sigma_{t_i}\sum_{k=1}^p h_i^{k+1}\varphi_{k+1}(h_i)\hat{\epsilon}_\theta^{(k)}(\hat{\boldsymbol{x}}_{\lambda_{t_{i-1}}}, \lambda_{t_{i-1}}) + \mathcal{O}(h^{p+2}). \tag{6}$$

where $\hat{\epsilon}_\theta^{(k)}$ denotes the $k$-th derivative of $\hat{\epsilon}_\theta$, and $\varphi_k(h)$ can be analytically computed [16]. The $\{a_m\}_{m=1}^p$ can be then determined by matching the coefficients between (4) and (6). In the following theorem, we show that UniC-$p$ has an order of accuracy $p + 1$ (see Appendix E.3 for detailed proof).

**Theorem 3.1** (The Order of Accuracy of UniC-$p$). *For any non-zero sequence $\{r_i\}_{i=1}^p$ and $h > 0$, define*

$$\boldsymbol{R}_p(h) = \begin{pmatrix} 1 & 1 & \cdots & 1 \\ r_1 h & r_2 h & \cdots & r_p h \\ \cdots & \cdots & \cdots & \cdots \\ (r_1 h)^{p-1} & (r_2 h)^{p-1} & \cdots & (r_p h)^{p-1} \end{pmatrix}.$$

*Let $\boldsymbol{\phi}_p(h) = (\phi_1(h), \cdots, \phi_p(h))^\top$ with $\phi_n(h) = h^n n!\varphi_{n+1}(h)$, where $\varphi_n(h)$ is defined by the recursive relation [16]:*

$$\varphi_{n+1}(h) = \frac{\varphi_n(h) - 1/n!}{h}, \quad \varphi_0(h) = e^h.$$

*For an increasing sequence $r_1 < r_2 < \cdots < r_p = 1$, suppose $\boldsymbol{a}_p := (a_1, \ldots, a_p)^\top$ satisfies,*

$$|\boldsymbol{R}_p(h_i)\boldsymbol{a}_p B(h_i) - \boldsymbol{\phi}_p(h_i)| = \mathcal{O}(h_i^{p+1}), \tag{7}$$

*where $|\cdot|$ denotes the $l_1$ norm for matrix. Then, under regularity conditions in Appendix E.2, UniC-$p$ of (4) will have $(p + 1)$-th order of accuracy.*

The monotonicity of $\{r_i\}_{i=1}^p$ ensures the invertibility of the Vandermonde matrix $\boldsymbol{R}_p$. Therefore, we can take $\boldsymbol{a}_p = \boldsymbol{R}_p^{-1}(h_i)\boldsymbol{\phi}_p(h_i)/B(h_i)$ as the coefficient vector for (4) for simplicity, where $B(h)$

can be any function of $h$ such that $B(h) = \mathcal{O}(h)$, for example $B_1(h) = h$, $B_2(h) = e^h - 1$. The detailed implementation of UniC is shown in Algorithm 1. Importantly, we circumvent the extra evaluation of $\epsilon_\theta(\tilde{\boldsymbol{x}}_{t_i}^c, t_i)$ by pushing $\epsilon_\theta(\tilde{\boldsymbol{x}}_{t_i}, t_i)$ into the buffer $Q$ instead of $\epsilon_\theta(\tilde{\boldsymbol{x}}_{t_i}^c, t_i)$. Taking full advantage of $\epsilon_\theta(\tilde{\boldsymbol{x}}_{t_i}, t_i)$ of previous results enables us to increase the order of accuracy without incurring significant increment of computation cost. This makes our method inherently different from the predictor-corrector methods in ODE literature [22], where the computational costs are doubled because an extra function evaluation on the corrected $\tilde{\boldsymbol{x}}_{t_i}^c$ is required for each step.

## 3.2 The Unified Predictor UniP-$p$

We find that the order of accuracy of UniC does not depend on the specific choice of the sequence $\{r_i\}_{i=1}^p$, which motivates us to design $p$-order unified predictor (UniP-$p$) which only leverages the previous $p$ data points by excluding $D_p$ in (4) since $D_p$ involves $\tilde{\boldsymbol{x}}_{t_i}$. The order of accuracy is guaranteed by the following corollary.

**Corollary 3.2** (The Order of Accuracy of UniP-$p$). *For an increasing sequence $r_1 < r_2 < \cdots < r_{p-1} < 1$, the solver given in* (4) *dropping the term $D_p$ and using coefficients that satisfies*

$$|\boldsymbol{R}_{p-1}(h_i)\boldsymbol{a}_{p-1}B(h_i) - \boldsymbol{\phi}_{p-1}(h_i)| = \mathcal{O}(h_i^p) \tag{8}$$

*has $p$-th order of accuracy.*

Due to the unified form of UniP and UniC, we can use UniP-$p$ as the implementation of the `Solver-p` in UniC-$p$ to obtain a new family of solvers called UniPC-$p$. Theorem 3.1 and Corollary 3.2 ensure that UniPC-$p$ can achieve $(p + 1)$-th order of accuracy. Moreover, under additional regularity conditions in Appendix D, based on Theorem 3.1 and Corollary 3.2, we show that the order of convergence of UniPC-$p$ reaches $p + 1$ (see Appendix D).

## 3.3 Comparison with Existing Methods

Here we discuss the connection and the difference between UniPC and previous methods. When $p = 1$, UniPC will reduce to DDIM [34]. Motivated by linear multistep approaches, PNDM [24] proposes to use pseudo numerical methods for DDIM, while our UniPC makes use of information in the ODE solution (2) and is specially designed for diffusion ODEs. DEIS [40] is built on exponential integrators in the time domain, where the integral cannot be analytically computed and explicit formulae for high-order solvers cannot be derived. By using the half log-SNR $\lambda$ [25, 26], it is shown that the application of integration-by-parts can simplify the integration of (2) and leads to explicit expansion of $\boldsymbol{x}_t$. DPM-Solver-2 [25] lies in our UniPC framework as UniP-1, where they assume $B(h) = e^h - 1$. We find through our numerical analysis that $B(h)$ can be any non-degenerate function such that $B(h) = \mathcal{O}(h)$. Furthermore, DPM-Solvers do not admit unified forms even for orders smaller than 3, which adds to the challenge of obtaining algorithms for higher orders. In contrast, our UniPC exploits the structure of exponential integrators w.r.t. half log-SNR and admits not only simple and analytical solutions for efficient computation but also unified formulations for easy implementation of any order.

## 3.4 Implementation

By setting $r_m = (\lambda_{t_{i-m-1}} - \lambda_{t_i})/h_i$, $m = 1, \ldots, p-1$, the UniPC-$p$ updates in a multistep manner, which reuses the previous evaluation results and proves to be empirically more efficient, especially for limited steps of model evaluation [11, 26], while singlestep methods might incur higher computation cost per step. Therefore, we use multistep UniPC in our experiments by default. The detailed algorithms for multistep UniPC and the proof of convergence can be found in Appendix B. For UniPC, the choices of $\{r_i\}_{i=1}^{p-1}$ determine different updating methods. If all the values are in $(0, 1]$, the UniPC will switch to singlestep. Notably, we find in experiments that our UniC consistently improves different updating methods. Besides, we find UniP-2 (8) and UniC-1 (7) degenerate to a simple equation where only a single $a_1$ is unknown, where we find $a_1 = 0.5$ can be a solution for both $B_1(h)$ and $B_2(h)$ (see Appendix F) independent of $h$. In Appendix C, we provide another variant of UniPC called UniPC$_v$ where the coefficients do not depend on $h$ for arbitrary order $p$.

In the conditional inference, guided sampling [14, 8] is often employed. Recent works [30, 26] find that thresholding data prediction models can boost the sampling quality and mitigate the problem of

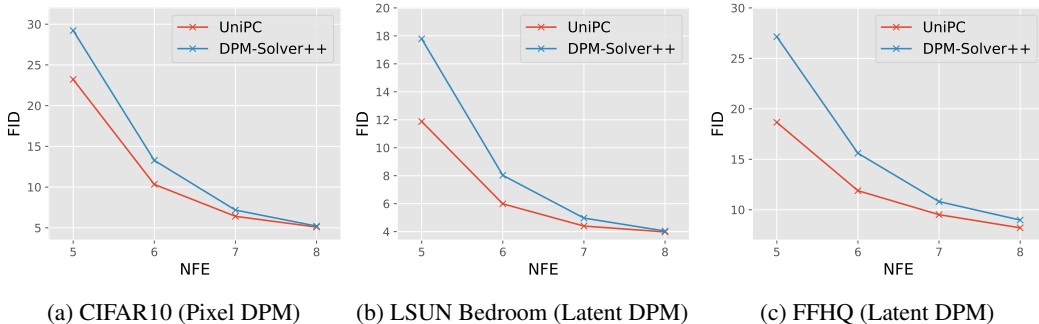

|  (a) CIFAR10 (Pixel DPM)  |  (b) LSUN Bedroom (Latent DPM)  |  (c) FFHQ (Latent DPM)  |

Figure 2: **Unconditional sampling results.** We compare our UniPC with DPM-Solver++ [25] on CIFAR10, LSUN Bedroom, and FFHQ. We report the FID↓ of the methods with different numbers of function evaluations (NFE). Experimental results demonstrate that our method is consistently better than previous ones on both pixel-space DPMs and latent-space DPMs, especially with extremely few steps. For more results, we recommend refering to Table 8-10 in Appendix G.

train-test mismatch. Our framework of UniPC can be easily adapted to the data prediction model, see Appendix A for algorithms and theoretical analysis. The detailed algorithms for multistep UniPC for data prediction are in Appendix B. Hence, UniPC with data prediction can achieve fast conditional sampling in extremely few steps through dynamic thresholding.

## 4    Experiments

In this section, we show that our UniPC can significantly improve the sampling quality through extensive experiments. Our experiments cover a wide range of datasets, where the image resolution ranges from $32\times32$ to $256\times256$. Apart from the standard image-space diffusion models [35, 8], we also conduct experiments on the recent prevailing stable-diffusion [29] trained on latent space. We will first present our main results in Section 4.1 and then provide a detailed analysis in Section 4.2.

### 4.1    Main Results

We start by demonstrating the effectiveness of our UniPC on both unconditional sampling and conditional sampling tasks, with extremely few model evaluations ($<10$ NFE). For the sake of clarity, we compare UniPC with the previous state-of-the-art method DPM-Solver++ [26]. We have also conducted experiments with other methods including DDIM [34], DPM-Solver [25], DEIS [40], and PNDM [24]. However, since some of these methods perform very unstable in few-step sampling, we leave their results in Section 4.2 and Appendix G.

**Unconditional sampling.** We first compare the unconditional sampling quality of different methods on CIFAR10 [21], FFHQ [18], and LSUN Bedroom [39]. The pre-trained diffusion models are from [35] and [29], including both pixel-space and latent-space diffusion models. The results are shown in Figure 2. For DPM-Solver++, we use the multistep 3-order version due to its better performance. For UniPC, we use a combination of UniP-3 and UniC-3, thus the order of accuracy is 4. As shown in Figure 3, we find that our UniPC consistently achieves better sampling quality than DPM-Solver++ on different datasets, especially with fewer NFE. Notably, compared with DPM-Solver++, our UniPC improves the FID by 6.0, 5.9, and 8.5 on CIFAR10, LSUN Bedroom, and FFHQ, respectively. These results clearly demonstrate that our UniPC can effectively improve the unconditional sampling quality with few function evaluations.

**Conditional sampling.** Conditional sampling is more useful since it allows user-defined input to control the synthesized image. To evaluate the conditional sampling performance of our UniPC, we conduct experiments on two widely used guided sampling settings, including classifier guidance and classifier-free guidance. For classifier guidance, we use the pixel-space diffusion model trained on ImageNet $256\times256$ [7] provided by [8]. Following DPM-Solver++, we use dynamic thresholding [30] to mitigate the gap between training and testing. The results are shown in Figure 3a and 3b, where we compare our UniPC with DPM-Solver++ [26] under different guidance scale ($s = 8.0/4.0$). For DPM-Solver++, we use the multistep 2-order version (2M), which achieves the best results

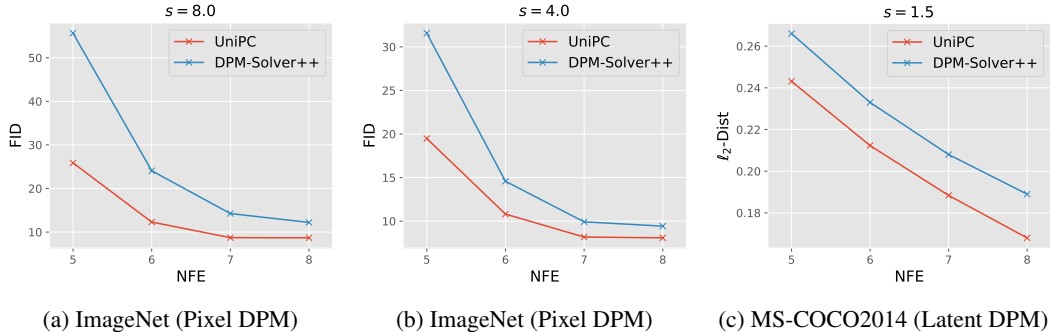

<table>
<tr><td>(a) ImageNet (Pixel DPM)</td><td>(b) ImageNet (Pixel DPM)</td><td>(c) MS-COCO2014 (Latent DPM)</td></tr>
</table>

Figure 3: **Conditional sampling results.** (a)(b) We compare the sample quality measured by FID↓ on ImageNet 256×256 with guidance scale $s = 8.0/4.0$; (c) We adopt the text-to-image model provided by stable-diffusion [29] to compare the convergence error, which is measured by the $l_2$ distance between the results of different methods and 1000-step DDIM. We show that our method outperforms previous ones with various guidance scales and NFE.

Table 1: **Ablation on the choice of** $B(h)$**.** We consider two implementations of $B(h)$ and also provide the performance of DPM-Solver++ [26] for comparison. The results are measured by the FID(↓) on CIFAR10 [21] and FFHQ [18]. We show that while the UniPC with both the two forms of $B(h)$ can outperform DPM-Solver++, $B_1(h)$ performs better at fewer sampling steps.

<table>
<tr><td colspan="5">(a) CIFAR10 (Pixel-space DPM)</td></tr>
<tr><td rowspan="2">Sampling Method</td><td colspan="4">NFE</td></tr>
<tr><td>5</td><td>6</td><td>8</td><td>10</td></tr>
<tr><td>DPM-Solver++ ([26])</td><td>29.22</td><td>13.28</td><td>5.21</td><td>4.03</td></tr>
<tr><td>UniPC ($B_1(h) = h$)</td><td>**23.22**</td><td>**10.33**</td><td>**5.10**</td><td>3.97</td></tr>
<tr><td>UniPC ($B_2(h) = e^h - 1$)</td><td>26.20</td><td>11.48</td><td>5.11</td><td>**3.87**</td></tr>
</table>

<table>
<tr><td colspan="5">(b) FFHQ (Latent-space DPM)</td></tr>
<tr><td rowspan="2">Sampling Method</td><td colspan="4">NFE</td></tr>
<tr><td>5</td><td>6</td><td>8</td><td>10</td></tr>
<tr><td>DPM-Solver++ ([26])</td><td>27.94</td><td>15.99</td><td>9.20</td><td>7.36</td></tr>
<tr><td>UniPC ($B_1(h) = h$)</td><td>**18.66**</td><td>**11.89**</td><td>**8.21**</td><td>**6.99**</td></tr>
<tr><td>UniPC ($B_2(h) = e^h - 1$)</td><td>21.66</td><td>13.21</td><td>8.63</td><td>7.20</td></tr>
</table>

according to the original paper. For our UniPC, we use UniP-2 and UniC-2. It can be seen that our UniPC generates samples with better quality and converges rather faster than other methods. For classifier-free guidance, we adopt the latent-space diffusion model provided by stable-diffusion [29] and set the guidance scale as 1.5 following their original paper. To obtain the input texts, we randomly sample 10K captions from MS-COCO2014 validation dataset [23]. As discussed in [26], the FID of the text-to-image saturates in <10 steps, possibly because the powerful decoder can generate good image samples from non-converged latent codes. Therefore, to examine how fast a method converges, we follow [26] to compute the $l_2$-distance between the generated latent code $x_0$ and the true solution $x_0^*$ (obtained by running a 999-step DDIM), *i.e.*, $\|x_0 - x_0^*\|_2/\sqrt{D}$, where $D$ is the dimension of the latent code. For each text input, we use the same initial value $x_T^*$ sampled from Gaussian distribution for all the compared methods. It can be seen in Figure 3c that our UniPC consistently has a lower $l_2$-distance than DPM-Solver++, which indicates that UniPC converges faster in guided sampling.

## 4.2 Analysis

In this section, we will provide more detailed analyses to further evaluate the effectiveness of UniPC.

**Ablation on the choice of** $B(h)$**.** In Section 3, we mentioned that $B(h)$ is set to be any non-zero function of $h$ that satisfies $B(h) = \mathcal{O}(h)$. We now investigate how the choice of $B(h)$ would affect the performance of our UniPC. Specifically, we test two simple forms: $B_1(h) = h$ and $B_2(h) = e^h - 1$ and the results are summarized in Table 1, where we also provide the performance of DPM-Solver++ [26] for reference. We show that UniPC with either implementation of $B(h)$ can outperform DPM-Solver++. When the NFE is extremely small (5~6), we observe that $B_1(h)$ consistently outperforms $B_2(h)$ by 1~3 in FID. On the other hand, as the NFE increases, the performance of $B_2(h)$ catches up and even surpasses $B_1(h)$ in some experiments (*e.g.*, on CIFAR10 and LSUN Bedroom). As for the guided sampling, we find $B_1(h)$ is worse than $B_2(h)$ consistently (see Appendix G for detailed results and discussions). These results also inspire us that our UniPC can be further improved by designing better $B(h)$, which we leave to future work.

Table 2: **Applying UniC to any solvers.** We show that UniC can be a plug-and-play component to boost the performance of both singlestep/multistep solvers with different orders. The sampling quality is measured by FID↓ on the CIFAR10 dataset.

| Sampling Method | Order | NFE | | | |
|---|---|---|---|---|---|
| | | 5 | 6 | 8 | 10 |
| DDIM ([34]) | 1 | 55.04 | 41.81 | 27.54 | 20.02 |
| + UniC (Ours) | 2 | 47.22 | 33.70 | 19.20 | 12.77 |
| DPM-Solver++(2M) ([26]) | 2 | 33.86 | 21.12 | 10.24 | 6.83 |
| + UniC (Ours) | 3 | 31.23 | 17.96 | 8.09 | 5.51 |
| DPM-Solver++ (3S) ([26]) | 3 | 51.49 | 38.83 | 11.98 | 6.46 |
| + UniC (Ours) | 4 | 50.62 | 24.59 | 10.32 | 5.50 |
| DPM-Solver++(3M) ([26]) | 3 | 29.22 | 13.28 | 5.21 | 4.03 |
| + UniC (Ours) | 4 | 25.50 | 11.72 | 5.04 | 3.90 |

Table 3: **Exploring the upper bound of UniC.** We compare the performance of UniC and UniC-oracle by applying them to the DPM Solver++. Note that the NFE of UniC-oracle is twice the number of sampling steps. Our results show that UniC still has room for improvement.

| Sampling Method | Sampling Steps | | | |
|---|---|---|---|---|
| | 5 | 6 | 8 | 10 |
| *LSUN Bedroom, Latent-space DPM* | | | | |
| DPM Solver++ ([26]) | 17.79 | 8.03 | 4.04 | 3.63 |
| + UniC | 13.79 | 6.53 | 3.98 | 3.52 |
| + UniC-oracle | 6.06 | 4.39 | 3.46 | 3.22 |
| *FFHQ, Latent-space DPM* | | | | |
| DPM Solver++ ([26]) | 27.15 | 15.60 | 8.98 | 7.39 |
| + UniC | 21.73 | 13.38 | 8.67 | 7.22 |
| + UniC-oracle | 15.29 | 11.25 | 8.33 | 7.03 |

**UniC for any order solver.** As shown in Algorithm 1, our UniC-$p$ can be applied after any $p$-order solver to increase the order of accuracy. To verify this, we perform experiments on a wide range of solvers. The existing solvers for DPM can be roughly categorized by the orders or the updating method (*i.e.*, singlestep or multistep). Since DPM-Solver++ [26] by design has both singlestep and multistep variants of 1∼3 orders, we apply our UniC to different versions of DPM-Solver++ to see whether UniC can bring improvements. The results are reported in Table 2, where the sampling quality is measured by FID↓ on CIFAR10 by sampling from a pixel-space DPM [35]. We also provide the order of accuracy of each baseline method without/with our UniC. Apart from the DDIM [34], which can be also viewed as 1-order singlestep DPM-Solver++, we consider another 3 variants of DPM-Solver++ including 2-order multistep (2M), 3-order singlestep (3S) and 3-order multistep (3M). It can be found that our UniC can increase the order of accuracy of the baseline methods by 1 and consistently improve the sampling quality for the solvers with different updating methods and orders.

**Exploring the upper bound of UniC.** According to Algorithm 1, our UniC works by leveraging the rough prediction $\tilde{x}_{t_i}$ as another data point to perform correction and increase the order of accuracy. Note that to make sure there is no extra NFE, we directly feed $\epsilon_\theta(\tilde{x}_{t_i}, t_i)$ to the next updating step instead of re-computing a $\epsilon_\theta(\tilde{x}_{t_i}^c, t_i)$ for the corrected $\tilde{x}_{t_i}^c$. Although the error caused by the misalignment between $\epsilon_\theta(\tilde{x}_{t_i}, t_i)$ and $\epsilon_\theta(\tilde{x}_{t_i}^c, t_i)$ has no influence on the order of accuracy (as proved in Appendix E.7), we are still interested in how this error will affect the performance. Therefore, we conduct experiments where we re-compute the $\epsilon_\theta(\tilde{x}_{t_i}^c, t_i)$ as the input for the next sampling step, which we name as "UniC-oracle". Due to the multiple function evaluations on each $t_i$ for both $\tilde{x}_{t_i}$ and $\tilde{x}_{t_i}^c$, the real NFE for UniC-oracle is twice as the standard UniC for the same sampling steps. However, UniC-oracle is very helpful to explore the upper bound of UniC, and thus can be used in pre-experiments to examine whether the corrector is potentially effective. We compare the performance of UniC and UniC-oracle in Table 3, where we apply them to the DPM Solver++ [26] on LSUN Bedroom [39] and FFHQ [18] datasets. We observe that the UniC-oracle can significantly improve the sampling quality over the baseline methods. Although the approximation error caused by the misalignment makes UniC worse than UniC-oracle, we find that UniC can still remarkably increases the sampling quality over the baselines, especially with few sampling steps.

**Customizing order schedule via UniPC.** Thanks to the unified analytical form of UniPC, we are able to investigate the performance of arbitrary-order solvers and customize the order schedule freely. As a first attempt, we conduct experiments on CIFAR10 with our UniPC, varying the order schedule (the order at each sampling step). Some results are listed in Table 4, where we test different order schedules with NFE=6/7 because the search space is not too big. Note that the order schedule in Table 4 represents the order of accuracy of the UniP, while the actual order is increased by 1 because of UniC. Our default order schedule follows the implementation of DPM-Solver++ [26], where lower-order solvers are used in the final few steps. Interestingly, we find some customized order schedules can yield better results, such as 123432 for NFE=6 and 1223334 for NFE=7. We also show that simply increasing the order as much as possible is harmful to the sampling quality.

**Sampling diversity.** Apart from the sampling quality, we are also interested in the diversity of the images generated by UniPC. In Table 6, we compare the sampling diversity of UniPC and

Table 4: **Customizing order schedule via UniPC.** We investigate different order schedules with UniPC and find that some customized order schedules behave better than the default settings, while simply increasing the order as much as possible is harmful to the sampling quality.

| _CIFAR10, NFE = 6_ | | | | |
| --- | --- | --- | --- | --- |
| Order Schedule | 123321 | 123432 | 123443 | 123456 |
| FID↓ | 10.33 | **9.03** | 11.23 | 22.98 |
| _CIFAR10, NFE = 7_ | | | | |
| Order Schedule | 1233321 | 1223334 | 1234321 | 1234567 |
| FID↓ | 6.41 | **6.29** | 7.24 | 60.99 |

Table 5: **Comparisons with more NFE.** We compare the sampling quality between UniPC and previous methods with 10-25 NFE on ImageNet 256×256 and show our UniPC still outperforms previous methods by a large margin.

| Sampling Method \ NFE | 10 | 15 | 20 | 25 |
| --- | --- | --- | --- | --- |
| DDIM ([34]) | 13.04 | 11.27 | 10.21 | 9.87 |
| DPM-Solver ([25]) | 114.62 | 44.05 | 20.33 | 9.84 |
| PNDM ([24]) | 99.80 | 37.59 | 15.50 | 11.54 |
| DEIS ([40]) | 19.12 | 11.37 | 10.08 | 9.75 |
| DPM-Solver++ ([26]) | 9.56 | 8.64 | 8.50 | 8.39 |
| UniPC (Ours) | **7.51** | **6.76** | **6.65** | **6.58** |

Table 6: **Comparisons of sampling diversity.** We compute the Inception Score (IS) on CIFAR10 [21] and find UniPC can generate more diverse samples than DPM-Solver++ [26].

| Inception Score (IS↑) | NFE | | | |
| --- | --- | --- | --- | --- |
| | 5 | 6 | 8 | 10 |
| DPM-Solver++ [26] | 7.27 | 8.62 | 9.52 | 9.69 |
| UniPC | **7.55** | **8.81** | **9.59** | **9.83** |

Table 7: **Comparisons of inference time.** We compare the inference time of sampling from a stable-diffusion [29] and find UniPC achieves a similar speed to DPM-Solver++ [26].

| Inference Time (s) | NFE | | |
| --- | --- | --- | --- |
| | 5 | 10 | 15 |
| DPM-Solver++ [26] | 0.48 | 0.77 | 1.07 |
| UniPC | 0.49 | 0.78 | 1.07 |

DPM-Solver++ [26], measured by the inception score (IS) on CIFAR10 dataset. We find that UniPC consistently generates more diverse samples in a variant number of function evaluations.

**Comparisons with more NFE.** To further evaluate the effectiveness of our UniPC, we also perform experiments with 10∼25 NFE. Specifically, we perform guided sampling on ImageNet 256×256 [7] with guidance scale 8.0 and compare our UniPC with more existing methods including DDIM, DPM-Solver, PNDM, DEIS, and DPM-Solver++. We summarize the results in Table 5, where some results of the previous methods are from [26]. The results clearly demonstrate that our UniPC surpasses previous methods by a large margin.

**Inference speed and memory.** We test the wall-clock time of UniPC by sampling from a stable-diffusion model [29] using a single NVIDIA RTX 3090 GPU and the results are shown in Table 7. We find the actual inference time of UniPC is very similar to DPM-Solver++ [26]. As for memory usage, it is only related to how many previous model outputs are stored. Therefore, our UniPC also costs similar memory to DPM-Solver++. For example, both UniPC-2 and DPM-Solver++(2M) cost about 6.3GB of memory when sampling from a stable-diffusion model.

**Visualizations.** We provide a qualitative comparison between our UniPC and previous methods with only 7 NFE, as shown in Figure 1. We use different methods to perform guided sampling with a guidance scale of 8.0 from a DPM trained on ImageNet 256×256. We find DEIS [40] tends to crash with extremely few steps, while the sample generated by DDIM [34] is relatively blurry. Compared with DPM-Solver++, the state-of-the-art method in guided sampling, UniPC can generate more plausible samples with better visual details. We further compare the sampling quality of our method UniPC and DPM-Solver++ using Stable-Diffusion-XL, a newly released model that can generate 1024 × 1024 images. The results in Figure 4 show that our method consistently generates more realistic images with fewer visual flaws.

**Limitations and broader impact.** Despite the effectiveness of UniPC, it still lags behind training-based methods such as [31]. How to further close the gap between training-free methods and training-based methods requires future efforts.

## 5   Conclusions

In this paper, we have proposed a new unified predictor-corrector framework named UniPC for the fast sampling of DPMs. Unlike previous methods, UniPC has a unified formulation for its two components (UniP and UniC) for any order. The universality of UniPC makes it possible to customize arbitrary order schedules and to improve the order of accuracy of off-the-shelf fast sampler via UniC.

| DPM-Solver++ [26] | UniPC (**Ours**) | DPM-Solver++ [26] | UniPC (**Ours**) |
|---|---|---|---|

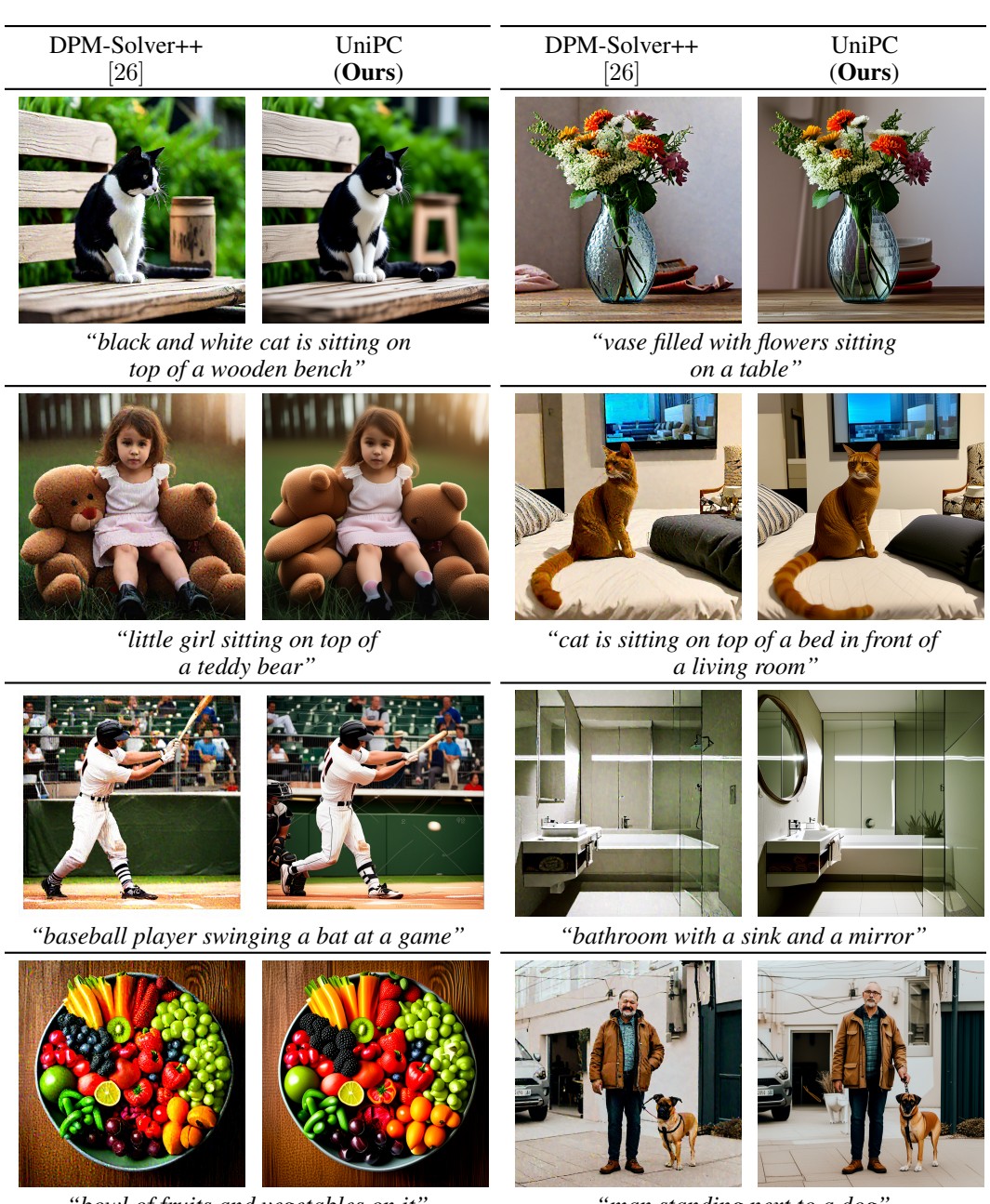

*"black and white cat is sitting on top of a wooden bench"*

*"vase filled with flowers sitting on a table"*

*"little girl sitting on top of a teddy bear"*

*"cat is sitting on top of a bed in front of a living room"*

*"baseball player swinging a bat at a game"*

*"bathroom with a sink and a mirror"*

*"bowl of fruits and vegetables on it"*

*"man standing next to a dog"*

Figure 4: Comparisons of text-to-image results between UniPC and DPM-Solver++[26]. Images are sampled from the newly released ***Stable-Diffusion-XL*** (1024×1024) using 15 NFE. We show that the images generated by DPM-Solver++ contain visible artifacts while UniPC consistently produces images with better quality. Please view the images in color and zoom in for easier comparison.

Extensive experiments have demonstrated the effectiveness of UniPC on unconditional/conditional sampling tasks with pixel-space/latent-space pre-trained DPMs. We have also discovered several directions where UniPC can be further improved, such as choosing a better $B(h)$, estimating a more accurate $\epsilon_\theta(\tilde{x}_{t_i}^c, t_i)$, and designing a better order schedule. We hope our attempt can inspire future work to further explore the fast sampling of DPMs in very few steps.

## Acknowledgments

This work was supported in part by the National Key Research and Development Program of China under Grant 2022ZD0160102, and in part by the National Natural Science Foundation of China under Grant 62321005, Grant 62336004, Grant 12271287 and Grant 62125603.

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

# A  UniPC for Data Prediction Model

## A.1  Comparison of data prediction and noise prediction

The data prediction model is a simple linear transformation of the noise prediction model, namely $\boldsymbol{x}_\theta = (\boldsymbol{x}_t - \sigma_t \boldsymbol{\epsilon}_\theta)/\alpha_t$ [19]. However, high-order solvers based on $\boldsymbol{x}_\theta$ and $\boldsymbol{\epsilon}_\theta$ are essentially different [30, 26]. As we shall see, the formulae of UniPC for the data prediction model differ from those for the noise prediction model. On the other hand, for image data, $\boldsymbol{x}_\theta$ is bounded in $[-1, 1]$, while $\boldsymbol{\epsilon}_\theta$ is generally unbounded and thus can push the sample out of the bound. Therefore, solvers for data prediction models are preferred, since thresholding method [30] can be directly applied and alleviate the "train-test mismatch" problem.

## A.2  Adapting UniPC to Data Prediction Model

As shown in the Proposition 4.1 of [26], for an initial value $\boldsymbol{x}_s$ at time $s > 0$, the solution at time $t \in [0, s]$ of diffusion ODEs is

$$\boldsymbol{x}_t = \frac{\sigma_t}{\sigma_s}\boldsymbol{x}_s + \sigma_t \int_{\lambda_s}^{\lambda_t} e^\lambda \hat{\boldsymbol{x}}_\theta(\hat{\boldsymbol{x}}_\lambda, \lambda)\mathrm{d}\lambda, \tag{9}$$

where we use the notation $\hat{\boldsymbol{x}}_\theta$ and $\hat{\boldsymbol{x}}_\lambda$ to denote changing from the domain of time $(t)$ to the domain of half log-SNR$(\lambda)$, *i.e.*, $\hat{\boldsymbol{x}}_\lambda := \boldsymbol{x}_{t_\lambda(\lambda)}$ and $\hat{\boldsymbol{x}}_\theta(\cdot, \lambda) := \boldsymbol{x}_\theta(\cdot, t_\lambda(\lambda))$. We are also able to adapt our UniPC to the data prediction model and utilize more information from previous data points. Recall that $h_i = \lambda_{t_i} - \lambda_{t_{i-1}}$. For any nonzero increasing sequence $r_1 < r_2 < \cdots < r_p = 1$, $\lambda_{s_m} = r_m h_i + \lambda_{t_{i-1}}, s_m = t_\lambda(\lambda_{s_m}), m = 1, \ldots, p$. The UniPC-$p$ is given by

$$\tilde{\boldsymbol{x}}_{t_i} = \frac{\sigma_{t_i}}{\sigma_{t_{i-1}}}\tilde{\boldsymbol{x}}_{t_{i-1}} + \alpha_{t_i}(1 - e^{-h_i})\boldsymbol{x}_\theta(\tilde{\boldsymbol{x}}_{t_{i-1}}, t_{i-1}) + \alpha_{t_i} B(h_i) \sum_{m=1}^{p-1} \frac{a_m}{r_m} D_m^x, \quad \text{(Predictor)} \tag{10}$$

$$\tilde{\boldsymbol{x}}_{t_i}^c = \frac{\sigma_{t_i}}{\sigma_{t_{i-1}}}\tilde{\boldsymbol{x}}_{t_{i-1}} + \alpha_{t_i}(1 - e^{-h_i})\boldsymbol{x}_\theta(\tilde{\boldsymbol{x}}_{t_{i-1}}, t_{i-1}) + \alpha_{t_i} B(h_i) \sum_{m=1}^{p} \frac{c_m}{r_m} D_m^x, \quad \text{(Corrector)} \tag{11}$$

where $D_m^x = \boldsymbol{x}_\theta(\tilde{\boldsymbol{x}}_{s_m}, s_m) - \boldsymbol{x}_\theta(\tilde{\boldsymbol{x}}_{t_{i-1}}, t_{i-1})$. Importantly, the corrector (UniC) can be also applied to any solver for the data prediction model that outputs $\tilde{\boldsymbol{x}}_{t_i}$. Denote $\boldsymbol{a}_p = (a_1, \cdots, a_{p-1})^\top$, $\boldsymbol{c}_p = (c_1, \cdots, c_p)^\top$. Let

$$\boldsymbol{g}_p(h) = (g_1(h), \cdots, g_p(h))^\top, \quad g_n(h) = h^n n! \psi_{n+1}(h), \tag{12}$$

where $\psi_n(h)$ is defined by the recursive relation $\psi_{n+1}(h) = \frac{1/n! - \psi_n(h)}{h}$, $\psi_0(h) = e^{-h}$, see Appendix E.4 for details. The order of accuracy of UniPC-$p$ for the data prediction model is given by the following proposition. The proof is in Appendix E.4.

**Proposition A.1** (Order of Accuracy of UniPC-$p$ for Data Prediction Model). *For an increasing sequence $r_1 < r_2 < \cdots < r_p = 1$, under regularity assumption E.3, assuming $0 \neq B(h) = \mathcal{O}(h)$,*

$$|\boldsymbol{R}_p(h_i)\boldsymbol{c}_p B(h_i) - \boldsymbol{g}_p(h_i)| = \mathcal{O}(h_i^{p+1}), \text{ and } |\boldsymbol{R}_{p-1}(h_i)\boldsymbol{a}_{p-1} B(h_i) - \boldsymbol{g}_{p-1}(h_i)| = \mathcal{O}(h_i^p), \tag{13}$$

*then the order of accuracy of UniPC-$p$ is $p + 1$.*

We list the algorithms for UniC-$p$ and UniP-$p$ for the data prediction model separately, see Algorithm 3 and Algorithm 4.

# B  Detailed Algorithms of Multistep UniPC

This section offers detailed algorithms with warming-up for multistep UniPC for noise prediction model (Algorithm 5,Algorithm 6) and for data prediction model (Algorithm 7,Algorithm 8).

# C  UniPC with varying coefficients (UniPC$_v$)

UniPC-$p$ (4) uses a vector $\boldsymbol{a}_p$ to match simultaneously all the coefficients of the derivatives. Alternatively, we can use $\{\boldsymbol{a}_{i,p}\}_{i=1}^p$ to match each order of derivatives separately and obtain a matrix of

---

**Algorithm 3** UniC-$p$ for data prediction model

---

**Require:** $\{r_i\}_{i=1}^{p-1}$, data prediction model $\boldsymbol{x}_\theta$, any $p$-order solver Solver-p, a buffer $Q = \{\boldsymbol{x}_\theta(\tilde{\boldsymbol{x}}_{t_{i-k}}, t_{i-k})\}_{k=1}^{p}$.

$h_i \leftarrow \lambda_{t_i} - \lambda_{t_{i-1}}, r_p \leftarrow 1, \tilde{\boldsymbol{x}}_{t_i} \leftarrow \texttt{Solver-p}(\tilde{\boldsymbol{x}}_{t_{i-1}}^c, Q)$

**for** $m = 1$ **to** $p$ **do**
    $s_m \leftarrow t_\lambda(r_m h_i + \lambda_{t_{i-1}})$
    $D_m^x \leftarrow \boldsymbol{x}_\theta(\tilde{\boldsymbol{x}}_{s_m}, s_m) - \boldsymbol{x}_\theta(\tilde{\boldsymbol{x}}_{t_{i-1}}, t_{i-1})$
**end for**

Compute $\boldsymbol{c}_p \leftarrow \boldsymbol{R}_p^{-1}(h_i)\boldsymbol{g}_p(h_i)/B(h_i)$, where $\boldsymbol{R}_p$ and $\boldsymbol{g}_p$ are as defined in Theorem 3.1 and (12)

$\tilde{\boldsymbol{x}}_{t_i}^c \leftarrow \frac{\sigma_{t_i}}{\sigma_{t_{i-1}}}\tilde{\boldsymbol{x}}_{t_{i-1}}^c + \alpha_{t_i}(1 - e^{-h_i})\boldsymbol{x}_\theta(\tilde{\boldsymbol{x}}_{t_{i-1}}, t_{i-1}) + \alpha_{t_i}B(h_i)\sum_{m=1}^{p} c_m D_m^x / r_m$

$Q \overset{\text{buffer}}{\leftarrow} \boldsymbol{x}_\theta(\tilde{\boldsymbol{x}}_{t_i}, t_i)$

**return:** $\tilde{\boldsymbol{x}}_{t_i}^c$

---

---

**Algorithm 4** UniP-$p$ for the data prediction model

---

**Require:** $\{r_i\}_{i=1}^{p-1}$, data prediction model $\boldsymbol{x}_\theta$, a buffer $Q = \{\boldsymbol{x}_\theta(\tilde{\boldsymbol{x}}_{t_{i-k}}, t_{i-k})\}_{k=1}^{p}$.

$h_i \leftarrow \lambda_{t_i} - \lambda_{t_{i-1}}$

**for** $m = 1$ **to** $p - 1$ **do**
    $s_m \leftarrow t_\lambda(r_m h_i + \lambda_{t_{i-1}})$
    $D_m^x \leftarrow \boldsymbol{x}_\theta(\tilde{\boldsymbol{x}}_{s_m}, s_m) - \boldsymbol{x}_\theta(\tilde{\boldsymbol{x}}_{t_{i-1}}, t_{i-1})$
**end for**

Compute $\boldsymbol{a}_{p-1} \leftarrow \boldsymbol{R}_{p-1}^{-1}(h_i)\boldsymbol{g}_{p-1}(h_i)/B(h_i)$, where $\boldsymbol{R}_{p-1}$ and $\boldsymbol{g}_{p-1}$ are as defined in Theorem 3.1 and (12)

$\tilde{\boldsymbol{x}}_{t_i} \leftarrow \frac{\sigma_{t_i}}{\sigma_{t_{i-1}}}\tilde{\boldsymbol{x}}_{t_{i-1}} + \alpha_{t_i}(1 - e^{-h_i})\boldsymbol{x}_\theta(\tilde{\boldsymbol{x}}_{t_{i-1}}, t_{i-1}) + \alpha_{t_i}B(h_i)\sum_{m=1}^{p-1} a_m D_m^x / r_m$

$Q \overset{\text{buffer}}{\leftarrow} \boldsymbol{x}_\theta(\tilde{\boldsymbol{x}}_{t_i}, t_i)$

**return:** $\tilde{\boldsymbol{x}}_{t_i}$

---

coefficients, *i.e.*, $\boldsymbol{A}_p = (\boldsymbol{a}_{1,p}, \cdots, \boldsymbol{a}_{p,p})^\top$. Consider

$$\tilde{\boldsymbol{x}}_{t_i} = \frac{\alpha_{t_i}}{\alpha_{t_{i-1}}}\tilde{\boldsymbol{x}}_{t_{i-1}} - \sigma_{t_i}(e^{h_i} - 1)\boldsymbol{\epsilon}_\theta(\tilde{\boldsymbol{x}}_{t_{i-1}}, t_{i-1}) - \sigma_{t_i}\sum_{n=1}^{p} h_i \varphi_{n+1}(h_i)\boldsymbol{a}_{n,p}^\top \boldsymbol{D}_p, \qquad (14)$$

where $\boldsymbol{D}_p = (D_1/r_1, \cdots, D_p/r_p)^\top$ and $D_m$ is defined in (5). The following theorem guarantees the order of accuracy of UniPC$_v$. The proof is deferred to Appendix E. The convergence order is investigated in Appendix D. Define

$$\boldsymbol{C}_p = \begin{pmatrix} 1 & 1 & \cdots & 1 \\ r_1/2! & r_2/2! & \cdots & r_p/2! \\ \vdots & \vdots & \ddots & \vdots \\ r_1^{p-1}/p! & r_2^{p-1}/p! & \cdots & r_p^{p-1}/p! \end{pmatrix}.$$

Let $\boldsymbol{I}_p$ be the $p$-dimensional identity matrix.

**Theorem C.1.** *Under the conditions of Theorem 3.1, if*

$$|\boldsymbol{C}_p \boldsymbol{A}_p - \boldsymbol{I}_p| = \mathcal{O}(h^p), \quad h = \max_{1 \le i \le p} h_i, \qquad (15)$$

*UniPC$_v$ is $(p+1)$-th order accurate.*

In fact, $\boldsymbol{C}_p$ is invertible. Note that $C_p$ is the product of a diagonal matrix and a Vandermonde matrix, namely

$$\boldsymbol{C}_p = \begin{pmatrix} 1 & & \\ & \ddots & \\ & & 1/p! \end{pmatrix} \begin{pmatrix} 1 & 1 & \cdots & 1 \\ r_1 & r_2 & \cdots & r_p \\ \vdots & \vdots & \ddots & \vdots \\ r_1^{p-1} & r_2^{p-1} & \cdots & r_p^{p-1} \end{pmatrix}, \quad r_1 < \cdots < r_p.$$

---

**Algorithm 5** Detailed implementation of multistep UniC-$p$

---

**Require:** initial value $\boldsymbol{x}_T$, time steps $\{t_i\}_{i=0}^M$ , noise prediction model $\boldsymbol{\epsilon}_\theta$, any $p$-order solver `Solver-p`

Denote $h_i := \lambda_{t_i} - \lambda_{t_{i-1}}$, for $i = 1, \cdots, M$.

$\tilde{\boldsymbol{x}}_{t_0} \leftarrow \boldsymbol{x}_T$, $\tilde{\boldsymbol{x}}_{t_0}^c \leftarrow \boldsymbol{x}_T$. Initialize an empty buffer $Q$.

$Q \overset{\text{buffer}}{\leftarrow} \boldsymbol{\epsilon}_\theta(\tilde{\boldsymbol{x}}_{t_0}, t_0)$

**for** $i = 1$ **to** $M$ **do**

    $p_i \leftarrow \min\{p, i\}$

    $\tilde{\boldsymbol{x}}_{t_i}^{(1)} \leftarrow \frac{\alpha_{t_i}}{\alpha_{t_{i-1}}} \tilde{\boldsymbol{x}}_{t_{i-1}}^c - \sigma_{t_i}(e^{h_i} - 1)\boldsymbol{\epsilon}_\theta(\tilde{\boldsymbol{x}}_{t_{i-1}}, t_{i-1})$

    $\tilde{\boldsymbol{x}}_{t_i} \leftarrow$ `Solver-p`$_i(\tilde{\boldsymbol{x}}_{t_{i-1}}^c, Q)$

    $r_{p_i} \leftarrow 1$

    $D_{p_i} \leftarrow \boldsymbol{\epsilon}_\theta(\tilde{\boldsymbol{x}}_{t_i}, t_i) - \boldsymbol{\epsilon}_\theta(\tilde{\boldsymbol{x}}_{t_{i-1}}, t_{i-1})$

    **for** $m = 2$ **to** $p_i$ **do**

        $r_{m-1} \leftarrow (\lambda_{t_{i-m}} - \lambda_{t_{i-1}})/h_i$

        $D_{m-1} \leftarrow \boldsymbol{\epsilon}_\theta(\tilde{\boldsymbol{x}}_{t_{i-m}}, t_{i-m}) - \boldsymbol{\epsilon}_\theta(\tilde{\boldsymbol{x}}_{t_{i-1}}, t_{i-1})$

    **end for**

    Compute $\boldsymbol{a}_{p_i} \leftarrow \boldsymbol{R}_{p_i}^{-1}(h_i)\boldsymbol{\phi}_{p_i}(h_i)/B(h_i)$, where $\boldsymbol{R}_{p_i}^{-1}(h_i)$ and $\boldsymbol{\phi}_{p_i}$ are as defined in Theorem 3.1.

    $\tilde{\boldsymbol{x}}_{t_i}^c \leftarrow \tilde{\boldsymbol{x}}_{t_i}^{(1)} - \sigma_{t_i} B(h_i) \sum_{m=1}^{p_i} a_m D_m / r_m$

    $Q \overset{\text{buffer}}{\leftarrow} \boldsymbol{\epsilon}_\theta(\tilde{\boldsymbol{x}}_{t_i}, t_i)$

**end for**

**return:** $\tilde{\boldsymbol{x}}_{t_M}$

---

Thus, we can simply take $\boldsymbol{A}_p = \boldsymbol{C}_p^{-1}$. The advantage of UniPC$_v$ is that $\boldsymbol{A}_p$ solely depends on $\{r_i\}_{i=1}^p$. The algorithm of UniPC$_v$ is to replace the updating formulae of UniPC (Algorithm 1 and Algorithm 2) by (14) with $\boldsymbol{A}_p = \boldsymbol{C}_p^{-1}$. See Appendix G for its performance.

# D  Order of Convergence

In this section, we shall show that under mild conditions the convergence order of UniP-$p$ is $p$ and the convergence order of UniPC-$p$ is $p + 1$ for either the noise prediction model or data prediction model. The proof is deferred to Appendix E.

**Definition D.1.** For time steps $\{t_i\}_{i=0}^M$, we say the order of convergence of a sampler for DPMs is $p$ if

$$|\tilde{\boldsymbol{x}}_{t_M} - \boldsymbol{x}_0| = \mathcal{O}(h^p).$$

We start by introducing some additional regularity assumptions.

**Assumption D.2.** The noise prediction model $\boldsymbol{\epsilon}_\theta(\boldsymbol{x}, s)$ is Lipschitz continuous w.r.t. $\boldsymbol{x}$ with Lipschitz constant $L$.

**Assumption D.3.** (1) $h = \max_{1 \leq i \leq M} h_i = \mathcal{O}(1/M)$ (2) For a constant $b > 0$, $\max_{1 \leq i \leq M} L\alpha_{t_{i-1}}/\alpha_{t_i} < b$, $b^{-1} < \alpha_{t_i} < b$, for all $1 \leq i \leq M$.

**Assumption D.4.** The starting values $\tilde{\boldsymbol{x}}_{t_i}$, $1 \leq i \leq k - 1$ satisfies for some positive constant $c_0$,

$$|\boldsymbol{x}_{t_i} - \tilde{\boldsymbol{x}}_{t_i}| \leq c_0 h^k, 1 \leq i \leq k - 1. \tag{16}$$

Assumption D.2 is common in the analysis of ODEs similar to (20). By Assumption D.2, we have $\boldsymbol{\epsilon}_\theta(\tilde{\boldsymbol{x}}_s, s) = \boldsymbol{\epsilon}_\theta(\boldsymbol{x}_s, s) + \mathcal{O}(\tilde{\boldsymbol{x}}_s - \boldsymbol{x}_s)$. Assumption D.3 assures that there is no significantly large step size and the signals are neither exploding nor degenerating. Assumption D.4 is common in the convergence analysis of multistep approaches [4].

The following Propositions D.5 and D.6 ensure the convergence order of UniP-$p$ and UniPC-$p$. For general `Solver-p` such as DDIM ($p = 1$), DPM-Solver/DPM-Solver++($p \leq 3$), UniC-$p$ can also increase the convergence order.

---

**Algorithm 6** Detailed implementation of multistep UniP-$p$

---

**Require:** initial value $\boldsymbol{x}_T$, time steps $\{t_i\}_{i=0}^M$, noise prediction model $\boldsymbol{\epsilon}_\theta$
Denote $h_i := \lambda_{t_i} - \lambda_{t_{i-1}}$, for $i = 1, \cdots, M$.
$\tilde{\boldsymbol{x}}_{t_0} \leftarrow \boldsymbol{x}_T$. Initialize an empty buffer $Q$.
$Q \overset{\text{buffer}}{\leftarrow} \boldsymbol{\epsilon}_\theta(\tilde{\boldsymbol{x}}_{t_0}, t_0)$
**for** $i = 1$ **to** $M$ **do**
   $p_i \leftarrow \min\{p, i\}$
   $\tilde{\boldsymbol{x}}_{t_i}^{(1)} \leftarrow \frac{\alpha_{t_i}}{\alpha_{t_{i-1}}} \tilde{\boldsymbol{x}}_{t_{i-1}} - \sigma_{t_i}(e^{h_i} - 1)\boldsymbol{\epsilon}_\theta(\tilde{\boldsymbol{x}}_{t_{i-1}}, t_{i-1})$
   **if** $p_i = 1$ **then**
      $\tilde{\boldsymbol{x}}_{t_i} \leftarrow \tilde{\boldsymbol{x}}_{t_i}^{(1)}$
      $Q \overset{\text{buffer}}{\leftarrow} \boldsymbol{\epsilon}_\theta(\tilde{\boldsymbol{x}}_{t_i}, t_i)$
      **continue**
   **end if**
   **for** $m = 2$ **to** $p_i$ **do**
      $r_{m-1} \leftarrow (\lambda_{t_{i-m}} - \lambda_{t_{i-1}})/h_i$
      $D_{m-1} \leftarrow \boldsymbol{\epsilon}_\theta(\tilde{\boldsymbol{x}}_{t_{i-m}}, t_{i-m}) - \boldsymbol{\epsilon}_\theta(\tilde{\boldsymbol{x}}_{t_{i-1}}, t_{i-1})$
   **end for**
   Compute $\boldsymbol{a}_{p_i-1} \leftarrow \boldsymbol{R}_{p_i-1}^{-1}(h_i)\boldsymbol{\phi}_{p_i-1}(h_i)/B(h_i)$, where $\boldsymbol{R}_{p_i-1}^{-1}(h_i)$ and $\boldsymbol{\phi}_{p_i-1}$ are as defined
   in Theorem 3.1.
   $\tilde{\boldsymbol{x}}_{t_i} \leftarrow \tilde{\boldsymbol{x}}_{t_i}^{(1)} - \sigma_{t_i}B(h_i)\sum_{m=1}^{p_i-1} a_m D_m/r_m$
   $Q \overset{\text{buffer}}{\leftarrow} \boldsymbol{\epsilon}_\theta(\tilde{\boldsymbol{x}}_{t_i}, t_i)$
**end for**
**return:** $\tilde{\boldsymbol{x}}_{t_M}$

---

**Proposition D.5.** *Under the conditions of Theorem 3.1, Assumptions D.2, D.3 , and D.4, the order of convergence of UniP-$p$ is $p$.*

**Proposition D.6.** *Under the conditions of Theorem 3.1, Assumptions D.2, D.3, and D.4 with $k = p$, the order of convergence of UniPC-$p$ is $p + 1$.*

*Remark* D.7. After a careful investigation of the proof of Proposition D.6 and Proposition D.5, we point out that for the singlestep updating when using UniPC-1 for the estimation of $\tilde{\boldsymbol{x}}_{s_m}$, $t_{i-1} < s_1, \cdots, s_m < t_i$, the order of convergence for UniPC-$p$ is $p + 1$ and the order of convergence for UniP-$p$ is $p$ for $p \leq 3$.

Following similar arguments of Proposition D.6, we find that UniC-$p$ can also increase the convergence order for general `Solver-p` for diffusion ODEs. The following is a direct corollary of Proposition D.6 which gives the order of convergence of UniPC$_v$-$p$.

**Corollary D.8.** *Under the conditions of Theorem C.1, Assumptions D.2, D.3, and D.4 with $k = p$, the order of convergence of UniPC$_v$-$p$ is $p + 1$.*

The order of convergence for the data prediction model follows analogous to Proposition D.5 and Proposition D.6 under slightly different assumptions:

**Assumption D.9.** The noise prediction model $\boldsymbol{x}_\theta(\boldsymbol{x}, s)$ is Lipschitz continuous w.r.t. $\boldsymbol{x}$ with Lipschitz constant $L$.

**Assumption D.10.** (1) $h = \max_{1 \leq i \leq M} h_i = \mathcal{O}(1/M)$ (2) For a constant $b > 0$, $\max_{1 \leq i \leq M} L\sigma_{t_{i-1}}/\sigma_{t_i} < b$, $b^{-1} < \sigma_{t_i} < \bar{b}$, for all $1 \leq i \leq M$.

We list the results of the order of convergence of UniP and UniPC for the data prediction model as corollaries to Propositions D.5 and D.6 and omit their proofs for simplicity. Interested readers can refer to the arguments in Appendix E for the noise prediction model and derive the detailed proofs for the data prediction model.

**Corollary D.11.** *Under the conditions of Proposition A.1, Assumptions D.9, D.10, and D.4 with $k = p$, the order of convergence of UniP-$p$ for data prediction model is $p$.*

**Corollary D.12.** *Under the conditions of Proposition A.1, Assumptions D.9, D.10, and D.4 with $k = p$, the order of convergence of UniPC-$p$ for data prediction model is $p + 1$.*

---

**Algorithm 7** Detailed implementation of multistep UniC-$p$ for data prediction model

---

**Require:** initial value $\boldsymbol{x}_T$, time steps $\{t_i\}_{i=0}^M$ , data prediction model $\boldsymbol{x}_\theta$, any p-order solver
Solver-p
Denote $h_i := \lambda_{t_i} - \lambda_{t_{i-1}}$, for $i = 1, \cdots, M$.
$\tilde{\boldsymbol{x}}_{t_0} \leftarrow \boldsymbol{x}_T, \tilde{\boldsymbol{x}}_{t_0}^c \leftarrow \boldsymbol{x}_T$. Initialize an empty buffer $Q$.
$Q \overset{\text{buffer}}{\leftarrow} \boldsymbol{x}_\theta(\tilde{\boldsymbol{x}}_{t_0}, t_0)$
**for** $i = 1$ **to** $M$ **do**
  $p_i \leftarrow \min\{p, i\}$
  $\tilde{\boldsymbol{x}}_{t_i}^{(1)} \leftarrow \frac{\sigma_{t_i}}{\sigma_{t_{i-1}}} \tilde{\boldsymbol{x}}_{t_{i-1}}^c + \alpha_{t_i}(1 - e^{-h_i})\boldsymbol{x}_\theta(\tilde{\boldsymbol{x}}_{t_{i-1}}, t_{i-1})$
  $\tilde{\boldsymbol{x}}_{t_i} \leftarrow \texttt{Solver-p}_i(\tilde{\boldsymbol{x}}_{t_{i-1}}^c, Q)$
  $r_{p_i} \leftarrow 1$
  $D_{p_i} \leftarrow \boldsymbol{x}_\theta(\tilde{\boldsymbol{x}}_{t_i}, t_i) - \boldsymbol{x}_\theta(\tilde{\boldsymbol{x}}_{t_{i-1}}, t_{i-1})$
  **for** $m = 2$ **to** $p_i$ **do**
    $r_{m-1} \leftarrow (\lambda_{t_{i-m}} - \lambda_{t_{i-1}})/h_i$
    $D_{m-1}^x \leftarrow \boldsymbol{x}_\theta(\tilde{\boldsymbol{x}}_{t_{i-m}}, t_{i-m}) - \boldsymbol{x}_\theta(\tilde{\boldsymbol{x}}_{t_{i-1}}, t_{i-1})$
  **end for**
  Compute $\boldsymbol{c}_{p_i} \leftarrow \boldsymbol{R}_{p_i}^{-1}(h_i)\boldsymbol{g}_{p_i}(h_i)/B(h_i)$, where $\boldsymbol{R}_{p_i}^{-1}(h_i)$ and $\boldsymbol{g}_{p_i}$ are as defined in Theorem 3.1
  and (12).
  $\tilde{\boldsymbol{x}}_{t_i}^c \leftarrow \tilde{\boldsymbol{x}}_{t_i}^{(1)} + \alpha_{t_i} B(h_i) \sum_{m=1}^{p_i} c_m D_m^x / r_m$
  $Q \overset{\text{buffer}}{\leftarrow} \boldsymbol{x}_\theta(\tilde{\boldsymbol{x}}_{t_i}, t_i)$
**end for**
**return:** $\tilde{\boldsymbol{x}}_{t_M}$

---

# E   Proofs

In this section, we provide preliminaries of numerical analysis, regularity assumption, and the proof of
Theorem 3.1 in the paper, the regularity assumption and the proof of Proposition A.1 in Appendix A
as well as detailed proofs for Theorem C.1 in Appendix C, for Proposition D.5 and Proposition D.6
in Appendix D. Through this section, we use $C$ and $C_i$ to denote sufficiently large positive constants
independent of $h$.

## E.1   Preliminaries

We begin with introducing preliminary results and concepts necessary for the proof of Theorem 3.1.

**Expansion of exponential integrator.** First, we obtain the Taylor expansion of (2), namely the
exponentially weighted integral. Define the $k$-th order derivative of $\hat{\boldsymbol{\epsilon}}_\theta(\hat{\boldsymbol{x}}_\lambda, \lambda)$ as $\hat{\boldsymbol{\epsilon}}_\theta^{(k)}(\hat{\boldsymbol{x}}_\lambda, \lambda) :=$
$d^k\hat{\boldsymbol{\epsilon}}_\theta(\hat{\boldsymbol{x}}_\lambda, \lambda)/d\lambda^k$. For $0 \leq t < s \leq T, r \in [0, 1]$, let $h := \lambda_t - \lambda_s, \lambda := \lambda_s + rh$. Assuming the
existence of total derivatives of $\hat{\boldsymbol{\epsilon}}_\theta^{(k)}(\hat{\boldsymbol{x}}_\lambda, \lambda), 0 \leq k \leq n$, we have the $n$-th order Taylor expansion of
$\hat{\boldsymbol{\epsilon}}_\theta(\hat{\boldsymbol{x}}_\lambda, \lambda)$ w.r.t. the half log-SNR $\lambda$:

$$\hat{\boldsymbol{\epsilon}}_\theta(\hat{\boldsymbol{x}}_\lambda, \lambda) = \sum_{k=0}^n \frac{r^k h^k}{k!} \hat{\boldsymbol{\epsilon}}_\theta^{(k)}(\hat{\boldsymbol{x}}_{\lambda_s}, \lambda_s) + \mathcal{O}(h^{n+1}). \tag{17}$$

Using the result of Taylor expansion of (17), the exponential integrator of (2) can be reduced to

$$\int_{\lambda_s}^{\lambda_t} e^{-\lambda}\hat{\boldsymbol{\epsilon}}_\theta(\hat{\boldsymbol{x}}_\lambda, \lambda)d\lambda = \frac{\sigma_t}{\alpha_t} \sum_{k=0}^n h^{k+1} \int_0^1 e^{\lambda_t - \lambda} \frac{r^k}{k!} dr \hat{\boldsymbol{\epsilon}}_\theta^{(k)}(\hat{\boldsymbol{x}}_{\lambda_s}, \lambda_s) + \mathcal{O}(h^{n+2})$$

$$= \frac{\sigma_t}{\alpha_t} \sum_{k=0}^n h^{k+1} \int_0^1 e^{(1-r)h} \frac{r^k}{k!} dr \hat{\boldsymbol{\epsilon}}_\theta^{(k)}(\hat{\boldsymbol{x}}_{\lambda_s}, \lambda_s) + \mathcal{O}(h^{n+2})$$

$$:= \frac{\sigma_t}{\alpha_t} \sum_{k=0}^n h^{k+1} \varphi_{k+1}(h) \hat{\boldsymbol{\epsilon}}_\theta^{(k)}(\hat{\boldsymbol{x}}_{\lambda_s}, \lambda_s) + \mathcal{O}(h^{n+2}), \tag{18}$$

---

**Algorithm 8** Detailed implementation of multistep UniP-$p$ for data prediction model

---

**Require:** initial value $\boldsymbol{x}_T$, time steps $\{t_i\}_{i=0}^M$, data prediction model $\boldsymbol{x}_\theta$

Denote $h_i := \lambda_{t_i} - \lambda_{t_{i-1}}$, for $i = 1, \cdots, M$.

$\tilde{\boldsymbol{x}}_{t_0} \leftarrow \boldsymbol{x}_T$. Initialize an empty buffer $Q$.

$Q \overset{\text{buffer}}{\Longleftarrow} \boldsymbol{x}_\theta(\tilde{\boldsymbol{x}}_{t_0}, t_0)$

**for** $i = 1$ **to** $M$ **do**

    $p_i \leftarrow \min\{p, i\}$

    $\tilde{\boldsymbol{x}}_{t_i}^{(1)} \leftarrow \frac{\sigma_{t_i}}{\sigma_{t_{i-1}}}\tilde{\boldsymbol{x}}_{t_{i-1}} + \alpha_{t_i}(1 - e^{-h_i})\boldsymbol{x}_\theta(\tilde{\boldsymbol{x}}_{t_{i-1}}, t_{i-1})$

    **if** $p_i = 1$ **then**

        $\tilde{\boldsymbol{x}}_{t_i} \leftarrow \tilde{\boldsymbol{x}}_{t_i}^{(1)}$

        $Q \overset{\text{buffer}}{\Longleftarrow} \boldsymbol{x}_\theta(\tilde{\boldsymbol{x}}_{t_i}, t_i)$

        **continue**

    **end if**

    **for** $m = 2$ **to** $p_i$ **do**

        $r_{m-1} \leftarrow (\lambda_{t_{i-m}} - \lambda_{t_{i-1}})/h_i$

        $D_{m-1}^x \leftarrow \boldsymbol{x}_\theta(\tilde{\boldsymbol{x}}_{t_{i-m}}, t_{i-m}) - \boldsymbol{x}_\theta(\tilde{\boldsymbol{x}}_{t_{i-1}}, t_{i-1})$

    **end for**

    Compute $\boldsymbol{a}_{p_i-1} \leftarrow \boldsymbol{R}_{p_i-1}^{-1}(h_i)\boldsymbol{g}_{p_i-1}(h_i)/B(h_i)$, where $\boldsymbol{R}_{p_i-1}^{-1}(h_i)$ and $\boldsymbol{g}_{p_i-1}$ are as defined in Theorem 3.1 and (12).

    $\tilde{\boldsymbol{x}}_{t_i} \leftarrow \tilde{\boldsymbol{x}}_{t_i}^{(1)} + \alpha_{t_i}B(h_i)\sum_{m=1}^{p_i-1} a_m D_m^x / r_m$

    $Q \overset{\text{buffer}}{\Longleftarrow} \boldsymbol{x}_\theta(\tilde{\boldsymbol{x}}_{t_i}, t_i)$

**end for**

**return:** $\tilde{\boldsymbol{x}}_{t_M}$

---

where $\varphi_{k+1}(h) = \int_0^1 e^{(1-r)h}\frac{r^k}{k!}\,\mathrm{d}r$ can be computed via the recurrence relation $\varphi_{k+1}(z) = (\varphi_k(z) - \varphi_k(0))/z$, $\varphi_k(0) = 1/k!$, and $\varphi_0(z) = e^z$ [16]. For example, the closed-forms of $\varphi_k(h)$ for $k = 1, 2, 3$ are

$$\varphi_1(h) = \frac{e^h - 1}{h}, \quad \varphi_2(h) = \frac{e^h - h - 1}{h^2}, \quad \varphi_3(h) = \frac{e^h - h^2/2 - h - 1}{h^3}.$$

**Order of accuracy.** In the following, we use the linear multistep method to illustrate the order of accuracy. Consider the ODE

$$y' = f(x, y), x \in [x_0, b], \quad y(x_0) = y_0. \tag{19}$$

We say that $f$ satisfies Lipschitz condition, if there exists $L > 0$ such that

$$|f(x, y_1) - f(x, y_2)| \le L|y_1 - y_2|, \forall y_1, y_2 \in \mathbb{R}. \tag{20}$$

Suppose $y(x)$ is the solution of Equation (19).

The $k$-order linear multistep method is given by

$$y_{n+k} = \sum_{i=0}^{k-1} \alpha_i y_{n+i} + h \sum_{i=0}^{k} \beta_i f(x_{n+i}, y_{n+i}), \tag{21}$$

where $y_{n+i}$ is the approximation of $y(x_{n+i})$, $x_{n+i} = x_n + ih$, $\alpha_i, \beta_i$ are constants, $|\alpha_0| + |\beta_0| > 0$.

**Definition E.1.** The local truncation error of (21) on $x_{n+k}$ is

$$T_{n+k} = y(x_{n+k}) - \sum_{i=0}^{k-1} \alpha_i y(x_{n+i}) - h \sum_{i=0}^{k} \beta_i f(x_{n+i}, y(x_{n+i})). \tag{22}$$

If $T_{n+k} = \mathcal{O}(h^{p+1})$, we say the order of accuracy of (21) is $p$.

The local truncation error describes the error caused by *one iteration* in the numerical analysis [22].

## E.2 Regularity Assumption

**Assumption E.2.** The total derivatives $\frac{\mathrm{d}^k \hat{\epsilon}_\theta(\hat{x}_\lambda, \lambda)}{\mathrm{d}\lambda_k}$, $k = 1, \cdots, p$ exist and are continuous.

Assumption E.2 is required for the Taylor expansion which is also regular in high-order numerical methods.

## E.3 Proof of Theorem 3.1 for arbitrary $p \geq 1$

We first give the local truncation error of UniC. Given $x_r$, $r \leq t_{i-1}$, are correct, with a slight abuse of notation, define

$$\bar{x}_{t_i} = \frac{\alpha_{t_i}}{\alpha_{t_{i-1}}} x_{t_{i-1}} - \sigma_{t_i}(e^{h_i} - 1)\epsilon_\theta(x_{t_{i-1}}, t_{i-1}) - \sigma_{t_i} B(h_i) \sum_{m=1}^{p-1} \frac{a_m}{r_m} (\hat{\epsilon}_\theta(\hat{x}_{\lambda_{s_m}}, \lambda_{s_m}) - \hat{\epsilon}_\theta(\hat{x}_{\lambda_{t_{i-1}}}, \lambda_{t_{i-1}}))$$
$$-\sigma_{t_i} B(h_i) \frac{a_p}{r_p}(\epsilon_\theta(\tilde{x}_{t_i}, t_i) - \hat{\epsilon}_\theta(\hat{x}_{\lambda_{t_{i-1}}}, \lambda_{t_{i-1}})).$$

Since `Solver-p` has order of accuracy $p$ and $B(h) = \mathcal{O}(h)$, under Assumption D.2, we have

$$\bar{x}_{t_i} = \frac{\alpha_{t_i}}{\alpha_{t_{i-1}}} x_{t_{i-1}} - \sigma_{t_i}(e^{h_i} - 1)\epsilon_\theta(x_{t_{i-1}}, t_{i-1}) - \sigma_{t_i} B(h_i) \sum_{m=1}^{p} \frac{a_m}{r_m}(\hat{\epsilon}_\theta(\hat{x}_{\lambda_{s_m}}, \lambda_{s_m})$$
$$- \hat{\epsilon}_\theta(\hat{x}_{\lambda_{t_{i-1}}}, \lambda_{t_{i-1}})) + \mathcal{O}(h^{p+2}). \tag{23}$$

Suppose $x_t$ is the solution of the diffusion ODE (1). Then, the local truncation error on $t_i$ is given by $|x_{t_i} - \bar{x}_{t_i}|$. Further, similar to Definition E.1 the order of accuracy of UniC is $l$, if there exists a sufficiently large positive constant $C$ such that

$$\max_{1 \leq i \leq M} |x_{t_i} - \bar{x}_{t_i}| \leq C h^{l+1}, \quad h = \max_{1 \leq i \leq M} h_i.$$

In the following, we shall show that the order of accuracy of UniC-$p$ is $p + 1$. Combining (2) and (17), we have

$$x_{t_i} = \frac{\alpha_{t_i}}{\alpha_{t_{i-1}}} x_{t_{i-1}} - \alpha_{t_i} \int_{\lambda_{t_{i-1}}}^{\lambda_{t_i}} e^{-\lambda} \hat{\epsilon}_\theta(\hat{x}_\lambda, \lambda) \mathrm{d}\lambda$$

$$= \frac{\alpha_{t_i}}{\alpha_{t_{i-1}}} x_{t_{i-1}} - \sigma_{t_i} \sum_{k=0}^{p} h_i^{k+1} \varphi_{k+1}(h_i) \hat{\epsilon}_\theta^{(k)}(\hat{x}_{\lambda_{t_{i-1}}}, \lambda_{t_{i-1}}) + \mathcal{O}(h^{p+2})$$

$$= \frac{\alpha_{t_i}}{\alpha_{t_{i-1}}} x_{t_{i-1}} - \sigma_{t_i}(e^{h_i} - 1)\epsilon_\theta(x_{t_{i-1}}, t_{i-1}) - \sigma_{t_i} \sum_{k=1}^{p} h_i^{k+1} \varphi_{k+1}(h_i) \hat{\epsilon}_\theta^{(k)}(\hat{x}_{\lambda_{t_{i-1}}}, \lambda_{t_{i-1}}) + \mathcal{O}(h^{p+2}).$$
$$\tag{24}$$

Let $\boldsymbol{R}_{p,k}^\top$ denote the $k$-th row of $\boldsymbol{R}_p$. By (7), we have $|\boldsymbol{a}_p^\top \boldsymbol{R}_{p,k}(h_i) - B^{-1}(h_i) h_i^k k! \varphi_{k+1}(h_i)| \leq C_0 B^{-1}(h_i) h^{p+1}$. Under Assumption E.2, by Taylor expansion, we obtain

$$\sum_{m=1}^{p} \frac{a_m}{r_m}(\hat{\epsilon}_\theta(\hat{x}_{\lambda_{s_m}}, \lambda_{s_m}) - \hat{\epsilon}_\theta(\hat{x}_{\lambda_{t_{i-1}}}, \lambda_{t_{i-1}}))$$

$$= \sum_{m=1}^{p} a_m \sum_{n=1}^{p} \frac{r_m^{n-1} h_i^n}{n!} \hat{\epsilon}_\theta^{(n)}(\hat{x}_{\lambda_{t_{i-1}}}, \lambda_{t_{i-1}}) + \mathcal{O}(h^{p+1})$$

$$= \sum_{n=1}^{p} \sum_{m=1}^{p} a_m \frac{r_m^{n-1} h_i^n}{n!} \hat{\epsilon}_\theta^{(n)}(\hat{x}_{\lambda_{t_{i-1}}}, \lambda_{t_{i-1}}) + \mathcal{O}(h^{p+1})$$

$$= \sum_{n=1}^{p} \frac{h_i^n}{n!} \boldsymbol{a}_p^\top \boldsymbol{R}_{p,n}(h_i) \hat{\epsilon}_\theta^{(n)}(\hat{x}_{\lambda_{t_{i-1}}}, \lambda_{t_{i-1}}) + \mathcal{O}(h^{p+1}).$$

Thus, we have

$$\left| \sum_{m=1}^{p} \frac{a_m}{r_m} (\hat{\boldsymbol{\epsilon}}_\theta(\hat{\boldsymbol{x}}_{\lambda_{s_m}}, \lambda_{s_m}) - \hat{\boldsymbol{\epsilon}}_\theta(\hat{\boldsymbol{x}}_{\lambda_{t_{i-1}}}, \lambda_{t_{i-1}})) - B(h_i)^{-1} \sum_{k=1}^{p} h_i^{k+1} \varphi_{k+1}(h_i) \hat{\boldsymbol{\epsilon}}_\theta^{(k)}(\hat{\boldsymbol{x}}_{\lambda_{t_{i-1}}}, \lambda_{t_{i-1}}) \right|$$

$$\leq C_1 (B(h_i)^{-1} h^{p+2} + h^{p+1}). \tag{25}$$

Combining (23), (24) and (25) given $B(h_i) = \mathcal{O}(h_i)$, we have

$$\max_{1 \leq i \leq M} |\boldsymbol{x}_{t_i} - \bar{\boldsymbol{x}}_{t_i}| = \Big| - \sigma_{t_i} \sum_{k=1}^{p} h_i^{k+1} \varphi_{k+1}(h_i) \hat{\boldsymbol{\epsilon}}_\theta^{(k)}(\hat{\boldsymbol{x}}_{\lambda_{t_{i-1}}}, \lambda_{t_{i-1}})$$

$$+ \sigma_{t_i} B(h_i) \sum_{m=1}^{p} \frac{a_m}{r_m} (\hat{\boldsymbol{\epsilon}}_\theta(\hat{\boldsymbol{x}}_{\lambda_{s_m}}, \lambda_{s_m}) - \hat{\boldsymbol{\epsilon}}_\theta(\hat{\boldsymbol{x}}_{\lambda_{t_{i-1}}}, \lambda_{t_{i-1}})) \Big| + \mathcal{O}(h^{p+2})$$

$$= \mathcal{O}(h^{p+2}). \tag{26}$$

Therefore, UniC-$p$ is of $(p+1)$-th order of accuracy. $\qquad\square$

### E.4 Proof of Proposition A.1

**Regularity Assumption**

**Assumption E.3.** The total derivatives $\frac{\mathrm{d}^k \hat{\boldsymbol{x}}_\theta(\hat{\boldsymbol{x}}_\lambda, \lambda)}{\mathrm{d}\lambda_k}$, $k = 1, \cdots, p$ exist and are continuous.

**Expansion of the Exponentially Weighted Integral.** First, we obtain the Taylor expansion of (9). Define the $k$-th order derivative of $\hat{\boldsymbol{x}}_\theta(\hat{\boldsymbol{x}}_\lambda, \lambda)$ as $\hat{\boldsymbol{x}}_\theta^{(k)}(\hat{\boldsymbol{x}}_\lambda, \lambda) := \mathrm{d}^k \hat{\boldsymbol{x}}_\theta(\hat{\boldsymbol{x}}_\lambda, \lambda)/\mathrm{d}\lambda^k$. For $0 \leq t < s \leq T$, $r \in [0,1]$, let $h := \lambda_t - \lambda_s$, $\lambda := \lambda_s + rh$. Assuming the existence of total derivatives of $\hat{\boldsymbol{x}}_\theta^{(k)}(\hat{\boldsymbol{x}}_\lambda, \lambda)$, $0 \leq k \leq n$, the $n$-th order Taylor expansion of $\hat{\boldsymbol{x}}_\theta(\hat{\boldsymbol{x}}_\lambda, \lambda)$ w.r.t. the half log-SNR $\lambda$ is:

$$\hat{\boldsymbol{x}}_\theta(\hat{\boldsymbol{x}}_\lambda, \lambda) = \sum_{k=0}^{n} \frac{r^k h^k}{k!} \hat{\boldsymbol{x}}_\theta^{(k)}(\hat{\boldsymbol{x}}_{\lambda_s}, \lambda_s) + \mathcal{O}(h^{n+1}). \tag{27}$$

Then, the exponential integrator of (9) can be reduced to

$$\int_{\lambda_s}^{\lambda_t} e^\lambda \hat{\boldsymbol{x}}_\theta(\hat{\boldsymbol{x}}_\lambda, \lambda) \mathrm{d}\lambda = \frac{\alpha_t}{\sigma_t} \sum_{k=0}^{n} h^{k+1} \int_0^1 e^{\lambda - \lambda_t} \frac{r^k}{k!} \mathrm{d}r \hat{\boldsymbol{x}}_\theta^{(k)}(\hat{\boldsymbol{x}}_{\lambda_s}, \lambda_s) + \mathcal{O}(h^{n+2})$$

$$= \frac{\alpha_t}{\sigma_t} \sum_{k=0}^{n} h^{k+1} \int_0^1 e^{(r-1)h} \frac{r^k}{k!} \mathrm{d}r \hat{\boldsymbol{x}}_\theta^{(k)}(\hat{\boldsymbol{x}}_{\lambda_s}, \lambda_s) + \mathcal{O}(h^{n+2})$$

$$:= \frac{\alpha_t}{\sigma_t} \sum_{k=0}^{n} h^{k+1} \psi_{k+1}(h) \hat{\boldsymbol{x}}_\theta^{(k)}(\hat{\boldsymbol{x}}_{\lambda_s}, \lambda_s) + \mathcal{O}(h^{n+2}), \tag{28}$$

where $\psi_{k+1}(h) = \int_0^1 e^{(r-1)h} \frac{r^k}{k!} \mathrm{d}r$ can be computed via the recurrence relation by integration-by-parts formula:

$$\psi_{k+1}(z) = \frac{1}{z} \int_0^1 \frac{r^k}{k!} \mathrm{d}e^{(r-1)z} = \frac{1}{z} \left( \frac{1}{k!} - \int_0^1 e^{(r-1)z} \frac{r^{k-1}}{(k-1)!} \mathrm{d}r \right) = \frac{1}{z} \left( \frac{1}{k!} - \psi_k(z) \right),$$

and $\psi_0(z) = e^{-z}$ [16]. For example, the closed-forms of $\psi_k(h)$ for $k = 1, 2, 3$ are

$$\psi_1(h) = \frac{1 - e^{-h}}{h}, \quad \psi_2(h) = \frac{h - 1 + e^{-h}}{h^2}, \quad \psi_3(h) = \frac{h^2/2 - h + 1 - e^{-h}}{h^3}.$$

**Proof.** In the subsequence analysis, we prove the order of accuracy of UniC for the data prediction model and the result of UniP for the data prediction model follows similarly. Suppose $\boldsymbol{x}_t$ is the

solution of the diffusion ODE (1). For the data prediction model, given $\boldsymbol{x}_r$, $r \le t_{i-1}$, are correct, with a slight abuse of notation, define

$$\bar{\boldsymbol{x}}_{t_i} = \frac{\sigma_{t_i}}{\sigma_{t_{i-1}}}\boldsymbol{x}_{t_{i-1}} + \alpha_{t_i}(1 - e^{-h_i})\boldsymbol{x}_\theta(\boldsymbol{x}_{t_{i-1}}, t_{i-1}) + \alpha_{t_i}B(h_i)\sum_{m=1}^{p-1}\frac{c_m}{r_m}(\hat{\boldsymbol{x}}_\theta(\hat{\boldsymbol{x}}_{\lambda_{s_m}}, \lambda_{s_m}) - \hat{\boldsymbol{x}}_\theta(\hat{\boldsymbol{x}}_{\lambda_{t_{i-1}}}, \lambda_{t_{i-1}}))$$
$$+\alpha_{t_i}B(h_i)\frac{c_p}{r_p}(\boldsymbol{x}_\theta(\tilde{\boldsymbol{x}}_{t_i}, t_i) - \hat{\boldsymbol{x}}_\theta(\hat{\boldsymbol{x}}_{\lambda_{t_{i-1}}}, \lambda_{t_{i-1}})).$$

Similar to (23), since `Solver-p` has order of accuracy $p$ and $B(h) = \mathcal{O}(h)$, under Assumption D.9, we have $\bar{\boldsymbol{x}}_{t_i} = \frac{\sigma_{t_i}}{\sigma_{t_{i-1}}}\boldsymbol{x}_{t_{i-1}} + \alpha_{t_i}(1-e^{-h_i})\boldsymbol{x}_\theta(\boldsymbol{x}_{t_{i-1}}, t_{i-1}) + \alpha_{t_i}B(h_i)\sum_{m=1}^{p}\frac{c_m}{r_m}(\hat{\boldsymbol{x}}_\theta(\hat{\boldsymbol{x}}_{\lambda_{s_m}}, \lambda_{s_m}) - \hat{\boldsymbol{x}}_\theta(\hat{\boldsymbol{x}}_{\lambda_{t_{i-1}}}, \lambda_{t_{i-1}})) + \mathcal{O}(h^{p+2})$. Then, the local truncation error on $t_i$ is given by $|\boldsymbol{x}_{t_i} - \bar{\boldsymbol{x}}_{t_i}|$. Further, similar to Definition E.1, we shall show that

$$\max_{1 \le i \le M}|\boldsymbol{x}_{t_i} - \bar{\boldsymbol{x}}_{t_i}| = \mathcal{O}(h^{p+2}), \quad h = \max_{1 \le i \le M}h_i.$$

Combining (9) and (28), we have

$$\boldsymbol{x}_{t_i} = \frac{\sigma_{t_i}}{\sigma_{t_{i-1}}}\boldsymbol{x}_{t_{i-1}} + \alpha_{t_i}(1 - e^{-h_i})\boldsymbol{\epsilon}_\theta(\boldsymbol{x}_{t_{i-1}}, t_{i-1}) + \alpha_{t_i}\sum_{k=1}^{p}h_i^{k+1}\psi_{k+1}(h_i)\hat{\boldsymbol{x}}_\theta^{(k)}(\hat{\boldsymbol{x}}_{\lambda_{t_{i-1}}}, \lambda_{t_{i-1}}) + \mathcal{O}(h^{p+2}).$$

Recall that $\boldsymbol{R}_{p,k}^\top$ is the $k$-th row of $\boldsymbol{R}_p$. By (13), we have $|\boldsymbol{c}_p^\top\boldsymbol{R}_{p,k}(h_i) - B^{-1}(h_i)h_i^k k!\psi_{k+1}(h_i)| \le CB^{-1}(h_i)h^{p+1}$ for some constant $C > 0$. Under Assumption E.3, by Taylor expansion, we obtain

$$\sum_{m=1}^{p}\frac{c_m}{r_m}(\hat{\boldsymbol{x}}_\theta(\hat{\boldsymbol{x}}_{\lambda_{s_m}}, \lambda_{s_m}) - \hat{\boldsymbol{x}}_\theta(\hat{\boldsymbol{x}}_{\lambda_{t_{i-1}}}, \lambda_{t_{i-1}}))$$
$$= \sum_{n=1}^{p}\sum_{m=1}^{p}c_m\frac{r_m^{n-1}h_i^n}{n!}\hat{\boldsymbol{x}}_\theta^{(n)}(\hat{\boldsymbol{x}}_{\lambda_{t_{i-1}}}, \lambda_{t_{i-1}}) + \mathcal{O}(h^{p+1})$$
$$= \sum_{n=1}^{p}\frac{h_i}{n!}\boldsymbol{c}_p^\top\boldsymbol{R}_{p,n}(h_i)\hat{\boldsymbol{x}}_\theta^{(n)}(\hat{\boldsymbol{x}}_{\lambda_{t_{i-1}}}, \lambda_{t_{i-1}}) + \mathcal{O}(h^{p+1}).$$

Thus, we have $\left|\sum_{m=1}^{p}\frac{c_m}{r_m}(\hat{\boldsymbol{x}}_\theta(\hat{\boldsymbol{x}}_{\lambda_{s_m}}, \lambda_{s_m}) - \hat{\boldsymbol{x}}_\theta(\hat{\boldsymbol{x}}_{\lambda_{t_{i-1}}}, \lambda_{t_{i-1}})) - B(h_i)^{-1}\sum_{k=1}^{p}h_i^{k+1}\psi_{k+1}(h_i)\hat{\boldsymbol{x}}_\theta^{(k)}(\hat{\boldsymbol{x}}_{\lambda_{t_{i-1}}}, \lambda_{t_{i-1}})\right| \le C_1(B(h_i)^{-1}h^{p+2} + h^{p+1})$. Given $0 \ne B(h_i) = \mathcal{O}(h_i)$, we have

$$\max_{1 \le i \le M}|\boldsymbol{x}_{t_i} - \bar{\boldsymbol{x}}_{t_i}| = \Big|\alpha_{t_i}\sum_{k=1}^{p}h_i^{k+1}\psi_{k+1}(h_i)\hat{\boldsymbol{x}}_\theta^{(k)}(\hat{\boldsymbol{x}}_{\lambda_{t_{i-1}}}, \lambda_{t_{i-1}})$$
$$- \alpha_{t_i}B(h_i)\sum_{m=1}^{p}\frac{c_m}{r_m}(\hat{\boldsymbol{x}}_\theta(\hat{\boldsymbol{x}}_{\lambda_{s_m}}, \lambda_{s_m}) - \hat{\boldsymbol{x}}_\theta(\hat{\boldsymbol{x}}_{\lambda_{t_{i-1}}}, \lambda_{t_{i-1}}))\Big| + \mathcal{O}(h^{p+2})$$
$$= \mathcal{O}(h^{p+2}).$$

Therefore, UniC-$p$ for the data prediction model is of $(p + 1)$-th order of accuracy. $\qquad\square$

### E.5  Proof of Theorem C.1

We first give the local truncation error of $\text{UniPC}_v$. Let $A_{m,n}$ denote the element of $\boldsymbol{A}_p$ on row $m$ and column $n$. Given $\boldsymbol{x}_r$, $r \le t_{i-1}$, are correct, with a slight abuse of notation, let

$$\check{\boldsymbol{x}}_{t_i} = \frac{\alpha_{t_i}}{\alpha_{t_{i-1}}}\boldsymbol{x}_{t_{i-1}} - \sigma_{t_i}(e^{h_i} - 1)\boldsymbol{\epsilon}_\theta(\boldsymbol{x}_{t_{i-1}}, t_{i-1})$$
$$- \sigma_{t_i}\sum_{n=1}^{p}h_i\varphi_{n+1}(h_i)\sum_{m=1}^{p-1}A_{m,n}(\hat{\boldsymbol{\epsilon}}_\theta(\hat{\boldsymbol{x}}_{\lambda_{s_m}}, \lambda_{s_m}) - \hat{\boldsymbol{\epsilon}}_\theta(\hat{\boldsymbol{x}}_{\lambda_{t_{i-1}}}, \lambda_{t_{i-1}}))/r_m$$
$$- \sigma_{t_i}\sum_{n=1}^{p}h_i\varphi_{n+1}(h_i)A_{p,n}(\boldsymbol{\epsilon}_\theta(\tilde{\boldsymbol{x}}_{t_i}, t_i) - \hat{\boldsymbol{\epsilon}}_\theta(\hat{\boldsymbol{x}}_{\lambda_{t_{i-1}}}, \lambda_{t_{i-1}}))/r_p.$$

Similar to (23), we have

$$\check{\boldsymbol{x}}_{t_i} = \frac{\alpha_{t_i}}{\alpha_{t_{i-1}}}\boldsymbol{x}_{t_{i-1}} - \sigma_{t_i}(e^{h_i} - 1)\boldsymbol{\epsilon}_\theta(\boldsymbol{x}_{t_{i-1}}, t_{i-1})$$

$$- \sigma_{t_i}\sum_{n=1}^{p}h_i\varphi_{n+1}(h_i)\sum_{m=1}^{p}A_{m,n}(\hat{\boldsymbol{\epsilon}}_\theta(\hat{\boldsymbol{x}}_{\lambda_{s_m}}, \lambda_{s_m}) - \hat{\boldsymbol{\epsilon}}_\theta(\hat{\boldsymbol{x}}_{\lambda_{t_{i-1}}}, \lambda_{t_{i-1}}))/r_m + \mathcal{O}(h^{p+2}). \quad (29)$$

Suppose $\boldsymbol{x}_t$ is the solution of the diffusion ODE (1) and the local truncation error on $t_i$ is given by $|\boldsymbol{x}_{t_i} - \check{\boldsymbol{x}}_{t_i}|$. Further, we shall show that the order of accuracy of UniPC$_v$-$p$ is $p + 1$, i.e.,

$$\max_{1 \le i \le M}|\boldsymbol{x}_{t_i} - \check{\boldsymbol{x}}_{t_i}| = \mathcal{O}(h^{p+2}), \quad h = \max_{1 \le i \le M}h_i.$$

By (24), we have

$$\boldsymbol{x}_{t_i} = \frac{\alpha_{t_i}}{\alpha_{t_{i-1}}}\boldsymbol{x}_{t_{i-1}} - \sigma_{t_i}(e^{h_i} - 1)\boldsymbol{\epsilon}_\theta(\boldsymbol{x}_{t_{i-1}}, t_{i-1}) - \sigma_{t_i}\sum_{k=1}^{p}h_i^{k+1}\varphi_{k+1}(h_i)\hat{\boldsymbol{\epsilon}}_\theta^{(k)}(\hat{\boldsymbol{x}}_{\lambda_{t_{i-1}}}, \lambda_{t_{i-1}}) + \mathcal{O}(h^{p+2}).$$

$$(30)$$

Under Assumption E.2, by Taylor expansion, we obtain

$$\sum_{k=1}^{p}h_i\varphi_{k+1}(h_i)\sum_{m=1}^{p}\frac{A_{m,k}}{r_m}(\hat{\boldsymbol{\epsilon}}_\theta(\hat{\boldsymbol{x}}_{\lambda_{s_m}}, \lambda_{s_m}) - \hat{\boldsymbol{\epsilon}}_\theta(\hat{\boldsymbol{x}}_{\lambda_{t_{i-1}}}, \lambda_{t_{i-1}}))$$

$$= \sum_{k=1}^{p}h_i\varphi_{k+1}(h_i)\sum_{m=1}^{p}A_{m,k}\sum_{n=1}^{p}\frac{r_m^{n-1}h_i^n}{n!}\hat{\boldsymbol{\epsilon}}_\theta^{(n)}(\hat{\boldsymbol{x}}_{\lambda_{t_{i-1}}}, \lambda_{t_{i-1}}) + \mathcal{O}(h^{p+2})$$

$$= \sum_{n=1}^{p}\sum_{k=1}^{p}\varphi_{k+1}(h_i)\sum_{m=1}^{p}A_{m,k}\frac{r_m^{n-1}h_i^{n+1}}{n!}\hat{\boldsymbol{\epsilon}}_\theta^{(n)}(\hat{\boldsymbol{x}}_{\lambda_{t_{i-1}}}, \lambda_{t_{i-1}}) + \mathcal{O}(h^{p+2})$$

$$= \sum_{n=1}^{p}\sum_{k=1}^{p}\varphi_{k+1}(h_i)h_i^{n+1}(\mathbf{1}(k = n) + \mathcal{O}(h^p))\hat{\boldsymbol{\epsilon}}_\theta^{(n)}(\hat{\boldsymbol{x}}_{\lambda_{t_{i-1}}}, \lambda_{t_{i-1}}) + \mathcal{O}(h^{p+2})$$

$$= \sum_{n=1}^{p}h_i^{n+1}\varphi_{n+1}(h_i)\hat{\boldsymbol{\epsilon}}_\theta^{(n)}(\hat{\boldsymbol{x}}_{\lambda_{t_{i-1}}}, \lambda_{t_{i-1}}) + \mathcal{O}(h^{p+2}).$$

Thus, we have

$$\max_{1 \le i \le M}|\boldsymbol{x}_{t_i} - \check{\boldsymbol{x}}_{t_i}| = \mathcal{O}(h^{p+2}).$$

Therefore, UniPC$_v$-$p$ is of $(p + 1)$-th order of accuracy. $\qquad\square$

### E.6 Proof of Proposition D.5

For UniP-$p$, we define

$$\bar{\boldsymbol{x}}_{t_i} = \frac{\alpha_{t_i}}{\alpha_{t_{i-1}}}\boldsymbol{x}_{t_{i-1}} - \sigma_{t_i}(e^{h_i} - 1)\boldsymbol{\epsilon}_\theta(\boldsymbol{x}_{t_{i-1}}, t_{i-1}) - \sigma_{t_i}B(h_i)\sum_{m=1}^{p-1}\frac{a_m}{r_m}(\hat{\boldsymbol{\epsilon}}_\theta(\hat{\boldsymbol{x}}_{\lambda_{s_m}}, \lambda_{s_m}) - \hat{\boldsymbol{\epsilon}}_\theta(\hat{\boldsymbol{x}}_{\lambda_{t_{i-1}}}, \lambda_{t_{i-1}})).$$

$$(31)$$

By Assumption D.2, we have $|\boldsymbol{\epsilon}_\theta(\tilde{\boldsymbol{x}}_s, s) - \boldsymbol{\epsilon}_\theta(\boldsymbol{x}_s, s)| \le L|\tilde{\boldsymbol{x}}_s - \boldsymbol{x}_s|$. Therefore, for sufficiently large constants $C, C_1 > 0$ depending on $\{a_m\}$ and $\{r_m\}$, we have

$$|\tilde{\boldsymbol{x}}_{t_i} - \bar{\boldsymbol{x}}_{t_i}| \le \Big(\frac{\alpha_{t_i}}{\alpha_{t_{i-1}}} + L\sigma_{t_i}(e^{h_i} - 1) + CLph\Big)|\tilde{\boldsymbol{x}}_{t_{i-1}} - \boldsymbol{x}_{t_{i-1}}| + CLh\sum_{m=1}^{p-1}|\tilde{\boldsymbol{x}}_{t_{i-m-1}} - \boldsymbol{x}_{t_{i-m-1}}|.$$

$$(32)$$

For simplicity, define $e_i = |\tilde{\boldsymbol{x}}_{t_i} - \boldsymbol{x}_{t_i}|$, $f_n = \max_{0 \le i \le n}|e_i|$. Using Theorem 3.1 and (32), we obtain

$$e_i \le \Big(\frac{\alpha_{t_i}}{\alpha_{t_{i-1}}} + L\sigma_{t_i}(e^{h_i} - 1) + CLph\Big)e_{i-1} + CLh\sum_{m=1}^{p-1}e_{i-1-m} + C_0h^{p+1}. \quad (33)$$

Let $\beta_i := \frac{\alpha_{t_i}}{\alpha_{t_{i-1}}} + L\sigma_{t_i}(e^{h_i} - 1) + CLph$. Then, it follows that

$$e_i \leq (CLph + \beta_i)f_{i-1} + C_0 h^{p+1}.$$

Since $\beta_i + CLph > 1$ for sufficiently small $h$, the right hand side is also a trivial bound for $f_{i-1}$, because $\alpha_t > 0$ is monotone decreasing thus $\alpha_{t_i} < \alpha_{t_{i-1}}$. We then have

$$f_i \leq (CLph + \beta_i)f_{i-1} + C_0 h^{p+1}. \tag{34}$$

Let $\sigma := \max_{1 \leq i \leq M} \sigma_{t_i} + 2Cp$. By elementary calculation, under Assumption D.3 we have

$$\prod_{i=p}^{M}(\beta_i + CLph) \leq C_1 \prod_{i=p}^{M}(\alpha_{t_i}/\alpha_{t_{i-1}} + L\sigma/M) = C_1 \frac{\alpha_{t_M}}{\alpha_{t_{p-1}}} \prod_{i=p}^{M}(1 + \frac{L\sigma\alpha_{t_{i-1}}}{\alpha_{t_i}M}) \leq C_2 e^{b\sigma}, \tag{35}$$

where $C_1$, $C_2$ are sufficiently large constants. Under Assumption D.4 with $k = p$, $f_{p-1} = \max_{0 \leq i \leq p-1}|\tilde{\boldsymbol{x}}_{t_i} - \boldsymbol{x}_{t_i}| \leq c_0 h^p$. Repeat the argument of (34), under Assumption D.3 by (35), there exists constant $C_3, C_4 > 0$ such that

$$f_M \leq C_2 e^{b\sigma} f_{p-1} + C_3 M h^{p+1} \leq C_4 h^p. \tag{36}$$

Therefore, the convergence order of UniP-$p$ is $p$. $\qquad\square$

### E.7 Proof of Proposition D.6

For the sake of clarity, we use $\bar{\boldsymbol{x}}_{t_i}^c = \frac{\alpha_{t_i}}{\alpha_{t_{i-1}}}\boldsymbol{x}_{t_{i-1}} - \sigma_{t_i}(e^{h_i} - 1)\boldsymbol{\epsilon}_\theta(\boldsymbol{x}_{t_{i-1}}, t_{i-1}) - \sigma_{t_i}B(h_i)\sum_{m=1}^{p}\frac{a_m}{r_m}(\hat{\boldsymbol{\epsilon}}_\theta(\hat{\boldsymbol{x}}_{\lambda_{s_m}}, \lambda_{s_m}) - \hat{\boldsymbol{\epsilon}}_\theta(\hat{\boldsymbol{x}}_{\lambda_{t_{i-1}}}, \lambda_{t_{i-1}}))$ for UniC-$p$ and use $\bar{\boldsymbol{x}}_{t_i}$ of (31) for UniP-$p$. Similar to (32), we have for $i \geq p$,

$$|\tilde{\boldsymbol{x}}_{t_i}^c - \bar{\boldsymbol{x}}_{t_i}^c| \leq \left(\frac{\alpha_{t_i}}{\alpha_{t_{i-1}}} + L\sigma_{t_i}(e^{h_i} - 1) + CLph\right)|\tilde{\boldsymbol{x}}_{t_{i-1}}^c - \boldsymbol{x}_{t_{i-1}}|$$

$$+ CLh\sum_{m=1}^{p-1}|\tilde{\boldsymbol{x}}_{t_{i-m-1}}^c - \boldsymbol{x}_{t_{i-m-1}}| + LhC_1|\tilde{\boldsymbol{x}}_{t_i} - \boldsymbol{x}_{t_i}|, \tag{37}$$

where for the oracle UniPC (see Section 4.2 for definition), the UniP-$p$ admits

$$|\tilde{\boldsymbol{x}}_{t_i} - \bar{\boldsymbol{x}}_{t_i}| \leq \left(\frac{\alpha_{t_i}}{\alpha_{t_{i-1}}} + L\sigma_{t_i}(e^{h_i} - 1) + CLph\right)|\tilde{\boldsymbol{x}}_{t_{i-1}}^c - \boldsymbol{x}_{t_{i-1}}|$$

$$+ CLh\sum_{m=1}^{p-1}|\tilde{\boldsymbol{x}}_{t_{i-m-1}}^c - \boldsymbol{x}_{t_{i-m-1}}|. \tag{38}$$

Define $e_i^c = |\tilde{\boldsymbol{x}}_{t_i}^c - \boldsymbol{x}_{t_i}|$, $f_n^c = \max_{0 \leq i \leq n}|e_i^c|$. Write $\beta_i := \frac{\alpha_{t_i}}{\alpha_{t_{i-1}}} + L\sigma_{t_i}(e^{h_i} - 1) + CLph$. Let $\sigma := \max_{1 \leq i \leq M} \sigma_{t_i} + 2Cp$. As shown in the proof of Theorem 3.1, $|\boldsymbol{x}_{t_i} - \bar{\boldsymbol{x}}_{t_i}^c| = \mathcal{O}(h^{p+2})$. Combining the local truncation error, it follows that

$$f_i^c \leq \beta_i f_{i-1}^c + CLhp f_{i-1}^c + LhC_1(\beta_i f_{i-1}^c + CLhp f_{i-1}^c + C_2 h^{p+1}) + C_0 h^{p+2}$$
$$\leq (1 + LhC_1)(\beta_i + CLhp)f_{i-1}^c + C_2 LC_1 h^{p+2} + C_0 h^{p+2}.$$

Repeating this argument, similar to (36), we have

$$|\tilde{\boldsymbol{x}}_{t_M}^c - \boldsymbol{x}_0| \leq f_M^c \leq \prod_{i=p}^{M}(1 + LhC_1)(\beta_i + CLhp)f_{p-1}^c + MC_2 h^{p+2}$$

$$\leq C_3 e^{LC_1 + b\sigma} f_{p-1}^c + MC_2 h^{p+2}. \tag{39}$$

For $0 \leq i < p$, we have

$$|\tilde{\boldsymbol{x}}_{t_i}^c - \bar{\boldsymbol{x}}_{t_i}| \leq \left(\frac{\alpha_{t_i}}{\alpha_{t_{i-1}}} + L\sigma_{t_i}(e^{h_i} - 1) + CLph\right)|\tilde{\boldsymbol{x}}_{t_{i-1}}^c - \boldsymbol{x}_{t_{i-1}}|$$

$$+ CLh\sum_{m=1}^{i-1}|\tilde{\boldsymbol{x}}_{t_{i-m-1}} - \boldsymbol{x}_{t_{i-m-1}}| + LhC_1|\tilde{\boldsymbol{x}}_{t_i} - \boldsymbol{x}_{t_i}|.$$

Therefore, under Assumption D.4 with $k = p$, we have for $0 \leq i \leq p - 1$

$$f_i^c \leq \beta_i f_{i-1}^c + c_0 C L p h^{p+1} + L h^{p+1} c_0 C_1.$$

Repeat this argument, we find for a constant $C_4 > 0$

$$f_{p-1}^c \leq C_4 h^{p+1}. \tag{40}$$

Combining (39) and (40), we have

$$|\tilde{\boldsymbol{x}}_{t_M}^c - \boldsymbol{x}_0| = \mathcal{O}(h^{p+1}).$$

For the non-oracle solver, for the step of UniP-$p$ we have :

$$|\tilde{\boldsymbol{x}}_{t_i} - \boldsymbol{x}_{t_i}| \leq \frac{\alpha_{t_i}}{\alpha_{t_{i-1}}} |\tilde{\boldsymbol{x}}_{t_{i-1}}^c - \boldsymbol{x}_{t_{i-1}}| + \left(CLph + L\sigma_{t_i}(e^{h_i} - 1)\right)|\tilde{\boldsymbol{x}}_{t_{i-1}} - \boldsymbol{x}_{t_{i-1}}|$$
$$+ CLh \sum_{m=1}^{p-1} |\tilde{\boldsymbol{x}}_{t_{i-m-1}} - \boldsymbol{x}_{t_{i-m-1}}| + C_2 h^{p+1}. \tag{41}$$

Let $e_n = |\tilde{\boldsymbol{x}}_{t_n} - \boldsymbol{x}_{t_n}|$, $f_i = \max_{1 \leq i \leq n} e_i$. Under Assumption D.4 with $k = p$, $f_{p-1} \leq c_0 h^p$. For $0 \leq i < p$, we have

$$|\tilde{\boldsymbol{x}}_{t_i}^c - \boldsymbol{x}_{t_i}| \leq \frac{\alpha_{t_i}}{\alpha_{t_{i-1}}} |\tilde{\boldsymbol{x}}_{t_{i-1}}^c - \boldsymbol{x}_{t_{i-1}}| + \left(CLph + L\sigma_{t_i}(e^{h_i} - 1)\right)|\tilde{\boldsymbol{x}}_{t_{i-1}} - \boldsymbol{x}_{t_{i-1}}|$$
$$+ CLh \sum_{m=1}^{i-1} |\tilde{\boldsymbol{x}}_{t_{i-m-1}} - \boldsymbol{x}_{t_{i-m-1}}| + LhC_1 |\tilde{\boldsymbol{x}}_{t_i} - \boldsymbol{x}_{t_i}| + C_0 h^{p+2}.$$

Similar to (40), we obtain $f_{p-1}^c = \mathcal{O}(h^{p+1})$ . Iterating (37) and (41), by Theorem 3.1 we have $|\tilde{\boldsymbol{x}}_{t_M}^c - \boldsymbol{x}_0| = \mathcal{O}(h^{p+1})$. $\qquad \square$

# F   Implementation Details

We now provide more details about our UniPC and the experiments.

## F.1   Implementation Details about UniPC

Our UniPC is implemented in a multistep manner by default, as is illustrated in Algorithm 5,6. In this case, the extra timesteps that are used to obtain the estimation of higher-order derivatives $\{s_m\}_{m=1}^{p-1}$ are set to be larger than $t_{i-1}$. In other words, $\{r_m\}_{m=1}^{p-1}$ are all negative. We have also found that the conditions of UniP-2 and UniC-1 degenerate to a simple equation where only a single $a_1$ is unknown. Specifically, considering (8) and (7), we find that

$$a_1 B(h) - \psi_1(h) = \mathcal{O}(h^2), \tag{42}$$

where

$$\psi_1(h) = h\varphi_2(h) = \frac{e^h - h - 1}{h} = \frac{1}{2}h + \mathcal{O}(h^2). \tag{43}$$

For $B_1(h) = h$, it is easy to show that when $a_1 = 0.5$,

$$a_1 B(h) - \psi_1(h) = \frac{1}{2}h - \frac{1}{2}h + \mathcal{O}(h^2) = \mathcal{O}(h^2). \tag{44}$$

For $B_2(h) = e^h - 1 = h + \mathcal{O}(h^2)$, the derivation is similar and $a_1 = 1/2$ also satisfies the condition. Therefore, we can directly set $a_1 = 1/2$ for UniP-2 and UniC-1 without solving the equation. For higher orders, the vector $\boldsymbol{a}_p$ is computed normally through the inverse of the $\boldsymbol{R}_p$ matrix.

To provide enough data points for high-order UniPC, we need a warming-up procedure in the first few steps, as is also used in previous multistep approaches [26] and is shown in Algorithm 5,6,. Since our UniC needs to compute $\epsilon_\theta(\tilde{\boldsymbol{x}}_{t_i}, t_i)$, *i.e.*, the model output at the current timestep $t_i$ to obtain the corrected result $\boldsymbol{x}_{t_i}^c$, performing our UniC at the last sampling step will introduce an extra function evaluation. Therefore, we do not use the corrector after the last execution of the predictor for fair comparisons.

Table 8: More unconditional sampling results on CIFAR10 [21].

| Sampling Method | NFE | | | | | |
|---|---|---|---|---|---|---|
| | 5 | 6 | 7 | 8 | 9 | 10 |
| DDIM [34] | 55.04 | 41.81 | 33.10 | 27.54 | 22.92 | 20.02 |
| DDIM + UniC-1 | 47.22 | 33.70 | 24.60 | 19.20 | 15.33 | 12.77 |
| DPM-Solver-3 [25] | 290.65 | 23.91 | 15.06 | 23.56 | 5.65 | 4.64 |
| DPM-Solver++(2M) [26] | 33.86 | 21.12 | 13.93 | 10.24 | 7.97 | 6.83 |
| DPM-Solver++(2M) + UniC-2 | 31.23 | 17.96 | 11.23 | 8.09 | 6.29 | 5.51 |
| DPM-Solver++(3M) [26] | 29.22 | 13.28 | 7.18 | 5.21 | 4.40 | 4.03 |
| DPM-Solver++(3M) + UniC-3 | 25.50 | 11.72 | 6.79 | 5.04 | 4.22 | 3.90 |
| UniPC-3-$B_1(h)$ | **23.22** | **10.33** | 6.41 | 5.10 | 4.29 | 3.97 |
| UniPC-3-$B_2(h)$ | 26.20 | 11.48 | 6.73 | 5.11 | 4.30 | **3.87** |
| UniPC$_v$-3 | 25.60 | 11.10 | **6.18** | **4.80** | **4.19** | 4.18 |

Table 9: More unconditional sampling results on FFHQ [18].

| Sampling Method | NFE | | | | | |
|---|---|---|---|---|---|---|
| | 5 | 6 | 7 | 8 | 9 | 10 |
| DDIM [34] | 58.23 | 44.40 | 34.52 | 28.06 | 23.52 | 19.72 |
| DDIM + UniC | 39.41 | 26.48 | 18.58 | 14.56 | 11.97 | 10.33 |
| DPM-Solver-3 [25] | 54.17 | 25.24 | 12.37 | 8.06 | 10.22 | 7.74 |
| DPM-Solver++(2M) [26] | 32.50 | 20.32 | 14.25 | 11.30 | 9.45 | 8.28 |
| DPM-Solver++(2M) + UniC | 24.20 | 14.92 | 10.82 | 9.11 | 8.01 | 7.39 |
| DPM-Solver++(3M) [26] | 27.15 | 15.60 | 10.81 | 8.98 | 7.89 | 7.39 |
| DPM-Solver++(3M) + UniC | 21.73 | 13.38 | 10.06 | 8.67 | 7.89 | 7.22 |
| UniPC-3-$B_1(h)$ | **18.66** | **11.89** | **9.51** | **8.21** | **7.62** | **6.99** |
| UniPC-3-$B_2(h)$ | 21.66 | 13.21 | 9.93 | 8.63 | 7.69 | 7.20 |

## F.2 Details about the experiments.

We now provide more details of our experiments. For unconditional sampling on CIFAR10 [21], we use the ScoreSDE [35] codebase and their pre-trained model, which is a continuous-time DDPM++ model [35]. More concretely, we use the `cifar10_ddpmpp_deep_continuous` config file, the same as the example provided by the official code of DPM-Solver [25]. To compute FID, we adopt the statistic file provided by ScoreSDE [35] codebase. For unconditional sampling on LSUN Bedroom [39] and FFHQ [18], we adopt the latent-space DPM provided by the stable-diffusion codebase [29]. Since there is no statistic file for these two datasets in the codebase, we compute the dataset statistic of FFHQ using the script in the library pytorch-fid, and borrow the statistic file of LSUN Bedroom from the guided-diffusion codebase [8]. For conditional sampling on pixel space, we implement our method in guided-diffusion codebase [8] and use the pre-trained checkpoint for ImageNet 256×256. For conditional sampling on latent space, we adopt the stable-diffusion codebase and use their `sd-v1-3.ckpt` checkpoint, which is pre-trained on LAION [32]. To obtain the text prompts, we randomly sample 10K captions from the MS-COCO2014 validation dataset [23]. We sample 10K random latent code $x_T^*$ for each caption and fix them when using different methods.

# G More Results

In this section, we will provide more detailed results, including both quantitative and qualitative results.

## G.1 More Quantitative Results

**Unconditional Sampling.** We start by demonstrating detailed results on CIFAR10 [21], which are shown in Table 8. The results of our proposed method are highlighted in gray. Apart from the results already illustrated in Figure 2, we also include the performance of the DPM-Solver [25]

Table 10: More unconditional results on LSUN [39].

| Sampling Method | NFE | | | | | |
|---|---|---|---|---|---|---|
| | 5 | 6 | 7 | 8 | 9 | 10 |
| DDIM [34] | 40.40 | 25.56 | 17.93 | 13.47 | 10.77 | 8.95 |
| DPM-Solver++(3M) [26] | 17.79 | 8.03 | 4.97 | 4.04 | 3.79 | 3.63 |
| DPM-Solver++(3M) + UniC | 13.79 | 6.53 | 4.58 | 3.98 | 3.69 | **3.52** |
| UniPC-3-$B_1(h)$ | **11.88** | **5.99** | **4.40** | **3.97** | **3.74** | 3.62 |
| UniPC-3-$B_2(h)$ | 13.60 | 6.44 | 4.47 | 3.91 | 3.76 | 3.54 |

Table 11: More conditional sampling results on ImageNet $256\times256$.

| Sampling Method | NFE | | | | | |
|---|---|---|---|---|---|---|
| | 5 | 6 | 7 | 8 | 9 | 10 |
| *guidance scale $s = 8.0$* | | | | | | |
| DDIM [34] | 38.11 | 25.31 | 17.71 | 14.50 | 11.51 | 10.34 |
| DEIS [40] | 83.80 | 67.73 | 54.91 | 44.91 | 37.84 | 31.84 |
| DPM-Solver++ [26] | 55.64 | 24.07 | 14.25 | 12.22 | 9.53 | 8.49 |
| UniPC-$B_1(h)$ | 68.69 | 34.76 | 22.79 | 20.45 | 17.44 | 16.02 |
| UniPC-$B_2(h)$ | **25.87** | **12.30** | **8.72** | **8.68** | **7.72** | **7.51** |
| *guidance scale $s = 4.0$* | | | | | | |
| DDIM [34] | 27.41 | 18.47 | 13.52 | 11.19 | 9.66 | 8.76 |
| DEIS [40] | 37.86 | 24.00 | 16.00 | 11.69 | 9.49 | 8.05 |
| DPM-Solver++ [26] | 31.57 | 14.58 | 9.92 | 9.43 | 7.98 | 7.51 |
| UniPC-$B_1(h)$ | 22.83 | 12.59 | 9.77 | 9.67 | 8.90 | 8.52 |
| UniPC-$B_2(h)$ | **19.48** | **10.81** | **8.18** | **8.09** | **7.49** | **7.31** |
| *guidance scale $s = 1.0$* | | | | | | |
| DDIM [34] | 36.08 | 27.02 | 21.38 | 17.59 | 15.20 | 13.40 |
| DPM-Solver++ [26] | 26.10 | 17.75 | 13.95 | 13.10 | 11.88 | 11.11 |
| UniPC-$B_2(h)$ | **22.22** | **15.79** | **12.72** | **12.28** | **11.30** | **10.84** |

and our UniPC$_v$ which has varying coefficients (see Appendix C for detailed description). Firstly, we show that DPM-Solver performs very unstable with extremely few steps: the FID of 5 NFE comes to 290.65, which means the solver crashes in this case. Besides, we also show that the DPM-Solver++(3M) [26] performs consistently better than DPM-Solver-3 [25], indicating the multistep method is more effective in few-step sampling setting. That is also why we tend to use DPM-Solver++ as our baseline method in all of our experiments. Table 8 also shows the comparisons between variants of our UniPC, such as UniPC with different instantiations of $B(h)$ and UniPC$_v$. Since the comparisons between $B_1(h)$ and $B_2(h)$ have already been discussed in Section 4.2, we now focus on the analysis of the performance of UniPC$_v$. We find UniPC$_v$ achieves the best sampling quality with 7 9 NFE, while cannot beat other variants of UniPC with 5,6,10 NFE. These results show that different variants of our UniPC may have different applications and we should select the most suitable one according to the budget of NFE. We also include more experimental results of the unconditional sampling results on FFHQ [18] in LSUN Bedroom [39], as summarized in Table 9 and Table 10. Note that there are fewer results for LSUN because we performed most of our experiments on FFHQ and CIFAR10 in the early stage. Nevertheless, the overall conclusions of the results on different datasets are aligned. To sum up, our results show (1) multistep methods behave better than singlestep ones when the NFE budget is extremely small. (2) our UniC can consistently improve the sampling quality of a wide range of off-the-shelf solvers. (3) selecting a proper variant of UniPC can yield better results in most cases.

**Conditional Sampling.** We also provide more results on guided sampling on ImageNet $256\times256$, as is shown in Table 11. We evaluate the performance of DEIS [40] and find it performs worse than both the DPM-Solver++ and our UniPC. Besides, we have compared the choice of $B(h)$ on guided

DDIM [34]

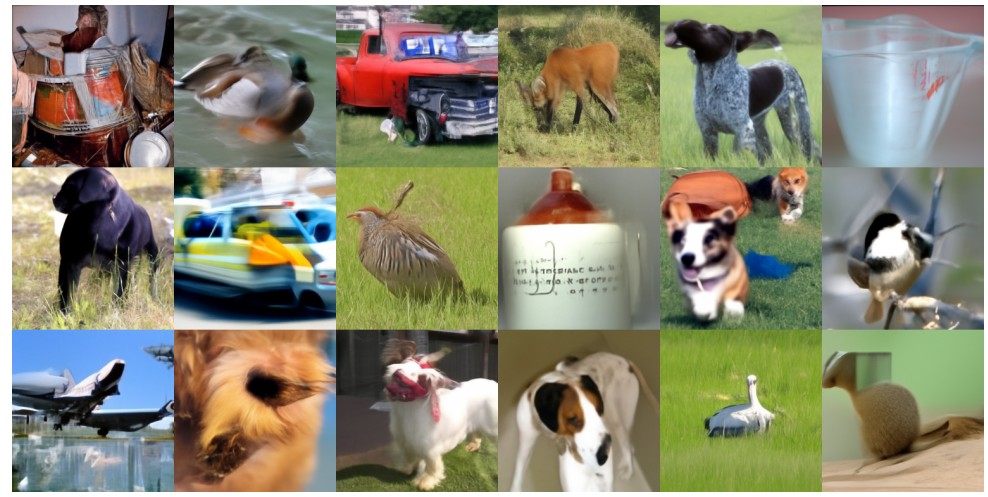

DPM-Solver++ [26]

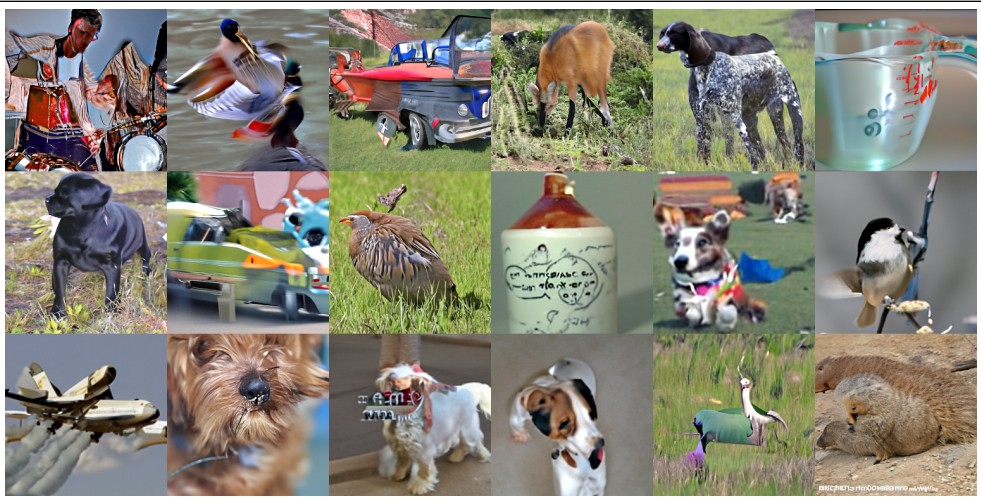

UniPC (**Ours**)

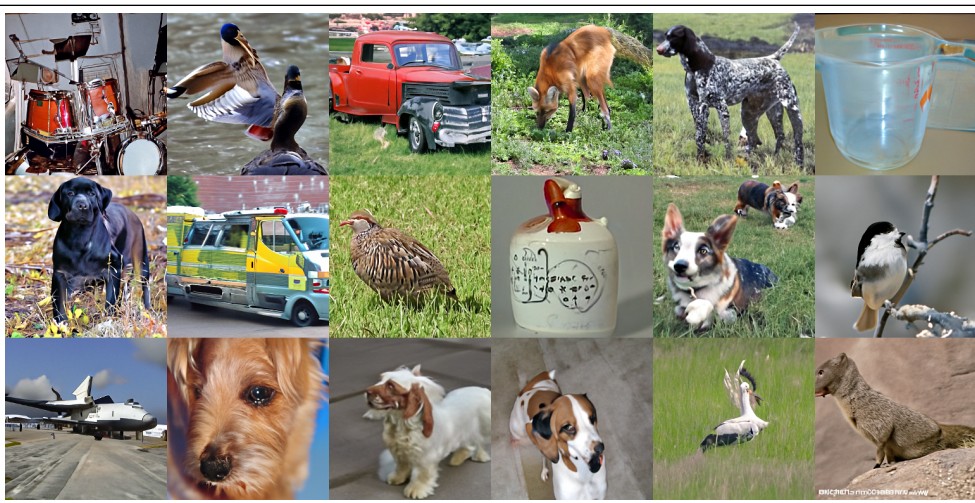

Figure 5: Comparisons between the images sampled from a DPM pre-trained on ImageNet256 × 256 using DDIM [34], DPM-Solver++ [26] and our UniPC with only 7 NFE.

sampling, where we find $B_2(h)$ significantly outperforms $B_1(h)$, perhaps because $B_1(h) = h$ is too simple and not suitable for the guided sampling. We have also added the results when the guidance scale $s = 1.0$. The results show that our method can achieve better sampling quality with both large and small guidance scales with few sampling steps.

### G.2 More Qualitative Results

We provide more visualizations to demonstrate the qualitative performance. First, we consider the class-conditioned guided sampling, *i.e.*, the conditional sampling on ImageNet [7] in Figure 5. Specifically, we compare the sampling quality of each method with only 7 NFE. Note that we randomly sample the class from the total 1000 classes in ImageNet for each image sample, but fix the initial noise for different methods.

