# OpenReview forum: "UniPC: A Unified Predictor-Corrector Framework for Fast Sampling of Diffusion Models"
_NeurIPS.cc/2023/Conference — NeurIPS 2023 poster_

### Official Review · Reviewer_xuvS · 2023-07-05

**Soundness:** 3 good
**Presentation:** 3 good
**Contribution:** 3 good
**Rating:** 7
**Confidence:** 2

**Summary:**

This paper propose a predictor-corrector method to accelerate the diffusion sampling process, where a corrector is proposed to correct the initial estimation of x_t using previous and current points. The experiments are conducted on imagenet and cifar10 both of which outperform existing efficient sampler at very few sampling steps.

**Strengths:**

1. Improving existing sampling method by a unified predictor-corrector solver is reasonable.
2. Adequate theoretical and empirical analysis of the proposed methods.
3. The paper is well organized and easy to understand.

**Weaknesses:**

N/A

**Questions:**

N/A

**Limitations:**

The authors adequately addressed the limitations

---

> ### Author Rebuttal · Authors · 2023-08-08
>
> We sincerely thank the reviewer for the positive comments on our work, especially the appreciation of our newly proposed unified framework UniPC, our adequate theoretical and empirical analysis, and our superior performance. We hope our work can open a new avenue for improving the sampling quality in the few-step sampling scenario via a predictor-corrector paradigm and thus promote the application of AIGC.

---

### Official Review · Reviewer_5Qcj · 2023-07-05

**Soundness:** 2 fair
**Presentation:** 2 fair
**Contribution:** 2 fair
**Rating:** 5
**Confidence:** 4

**Summary:**

This paper develops a unified corrector (UniC) that can be applied after any existing DPM sampler to increase the order of accuracy without extra model evaluations, and derive a unified predictor (UniP) that supports arbitrary order as a byproduct. Combining UniP and UniC,  a unified predictor-corrector framework called UniPC for the fast sampling of DPMs  has a unified analytical form for any order and can significantly improve the sampling quality over previous methods, especially in extremely few steps.

**Strengths:**

On both unconditional and conditional sampling using pixel-space and latent-space DPMs. UniPC can achieve 3.87 FID on CIFAR10 (unconditional) and 7.51 FID on ImageNet 256 256 (conditional) with only 10 function evaluations.

**Weaknesses:**

They claim many times on unified or model-agnostic but no experiments support their claim, the method only tested on DPM. Also the writing make me lost  many times when reading the paper, I suggest you rewrite your motivation part by which technical issue in DPM you try to solve, why your method works.  The theory on faster convergence property is also confused on accuracy, is it really useful?  how to  theoretically guarantee on the accuracy.

**Questions:**

They claim many times on unified or model-agnostic but no experiments support their claim, the method only tested on DPM. Also the writing make me lost  many times when reading the paper, I suggest you rewrite your motivation part by which technical issue in DPM you try to solve, why your method works.  The theory on faster convergence property is also confused on accuracy, is it really useful?  how to  theoretically guarantee on the accuracy.

**Limitations:**

See Weaknesses.

---

> ### Author Rebuttal · Authors · 2023-08-08
>
> We sincerely thank the reviewer for the positive comments on our work! We address the questions and clarify the issues accordingly as described below.
>
> **Q1: About model-agnostic**
>
> **[Reply]** Sorry for the confusion. By “model-agnostic” we mean that (1): our UniC can be applied after any existing solver, which is proved in Table 2; (2): like other training-free samplers, our method can be used to sample from any off-the-shelf DPMs, (pixel/latent space DPMs of different resolutions, see Figure 2,3). We will clarify this in the revised paper.
>
> **Q2: About the motivation**
>
> **[Reply]** Thanks for your suggestion. Our main motivation is that existing solvers for DPM often suffer from large accumulative error when sampling with <10 NFEs, due to the large step size. To overcome this issue, we propose to use a corrector than re-use the current point to reduce the error and increase the order of accuracy (which are proved theoretically and empirically). We will add the above discussion to make our motivation clearer.
>
> **Q3: About the accuracy**
>
> **[Reply]** The theoretical accuracy (i.e., order of accuracy/convergence order) is guaranteed by the Theorem 3.1 (which is proved in Appendix E.3). We have also conducted extensive experiments to further prove that our method can indeed improve the sampling quality of the DPMs. We will also improve our writing to highlight the novelty and usefulness of our newly proposed UniPC.

---

### Official Review · Reviewer_TtDL · 2023-07-07

**Soundness:** 2 fair
**Presentation:** 2 fair
**Contribution:** 2 fair
**Rating:** 3
**Confidence:** 4

**Summary:**

This paper presents a universal predictor-corrector method for faster sampling of diffusion models, with model retraining. The key idea is to further include the current point along with previous $p$ points while estimating the data point by adding a correction step. It shows that this model can achieve an order of accuracy $p+1$. Experiments on several image datasets show that this proposed method achieves strong performance in comparison to prior SOTA methods.



**Strengths:**

The paper is generally written clearly

The problem of accelerating diffusion model is critical to solve

Reasonably good performance achieved

**Weaknesses:**

**Limited novelty and design**:
- The idea proposed here is marginal in the sense that at each step, the current estimate is additionally used along with some previous points  (the latte is the same as [25,34,40]); Given the sequential nature, the current point is actually already used by previous methods [25,34,40] in the next steps (with a step delay), whilst it is just that this point is used for more times. No extra information is used whilst also no clear understand on why this extra use of the current point can lead to increase in the order of accuracy.

- Besides, in general, there are two families of diffusion models: DDPM (e.g., [A, 34]) and SGM (e.g., [35]). And the DDPM series are often easier to speed up due to adopting a variance preserving (VP) process, in contrast to SGM’s variance exploding (VE) process. It is not clear which family this proposed method is focused on, or both?

- It looks the step of 10 is a milestone. Is there any relationship or theoretical implication of this proposed method on tackling this issue?

- Also, the addition operations introduce some complex parameter, which adds further burden to the model tuning process. It is unclear if these parameters are general for different datasets.

**Limited results gain**
- Whilst the numeral results such as FID looks good in comparison, when it comes to the visual examples in supplementary, the generated images are of similar quality as previous methods such as DPM-Solver++. It is known that numerical metrics are some limited in the visual perception evaluation. Overall, the generation performance gain is some limited, not as convincing as what claimed.

- It is some inconsistent when evaluating UniC/UniP/UniPC. How is this selection done?

- If I am not missing, there is no exact evaluation on the benefit of using the current point in estimation, which is the key design of the whole  model.

**References**:
- [A] Jonathan et al. Denoising Diffusion Probabilistic Models. NeurIPS 2020

**Questions:**

Please see the weaknesses above

---

> ### Author Rebuttal · Authors · 2023-08-08
>
> We thank the reviewer for the comments. We address the questions and clarify the issues accordingly as described below.
>
> **Q1: About the novelty and design**
>
> **[Reply]**
> - As discussed in Section 3.1 and proved in Appendix E.3, our UniC can indeed increase the order of accuracy of the sampling procedure. We would like to clarify that our usage of the current point is better than [25, 34, 40] because we reduce the error of the current point via a corrector step, while in previous methods there are more accumulative errors. We will add more discussion about how our method works in the revised paper.
> - Thanks for your advice. Our method is designed for the variance preserving (VP) diffusion models, similar to DPM-Solver. VP diffusion models are also more useful in practice, such as the Stable-Diffusion, DeepFloyd-IF, etc. We will clarify this in our paper.
> - We think that the proposed corrector (UniC) is the main reason why our method works better than previous methods when NFE<10. Existing methods would suffer from large accumulative errors with large step sizes. However, our method can mitigate this issue because we can obtain a better estimation for the current point due to the extra corrector step. Please also refer to Table.2, where we show that UniC can boost the performance of a variety of existing solvers.
> - Sorry for the confusion. Our method is totally training-free, containing no extra learnable parameters. Our method can serve as a drop-in replacement of existing samplers of diffusion models and accelerate the sampling process.
>
> **Q2: About the results gain**
>
> **[Reply]**
> - We would need to focus more on the detailed structure of the images when comparing the qualitative results. For example in Figure 4, it can be easily found that our UniPC generates more realistic images than DPM-Solver++ (some blurry or broken regions can be easily observed in the images generated by DPM-Solver++). We also encourage the reviewer to have a look at the results of sampling from a larger model _**Stable-Diffusion-XL**_ in the attached one-page PDF, where we show that UniPC can generate more realistic images than the baseline method.
> - For most of the experiments, we use the UniPC (which is the combination of UniP and UniC) to compare with other methods. In the ablation study of UniC (see Table 2), we directly add the UniC after a variety of existing solvers demonstrate the UniC can be a plug-and-play module to improve sampling quality.
> - Please refer to Table 2, where we clearly show that using the current point in the corrector step can consistently boost performance.

---

> ### Comment · Area_Chair_t2zM · 2023-08-18
> **Could you look at the authors' response?**
>
> Dear reviewer:
>
> Please look at the authors' response and other reviewer' comments. Thanks!

---

### Official Review · Reviewer_vcSP · 2023-07-10

**Soundness:** 3 good
**Presentation:** 3 good
**Contribution:** 3 good
**Rating:** 5
**Confidence:** 3

**Summary:**

In this paper, the authors present a novel sampling solver called UniPC for diffusion models. UniPC consists of two parts: UniC and UniP, where UniC corrects the estimation with prediction of current timestep and it can be applied other sampling solvers and UniP is a special case of UniC and share the similar form for sampling. Compared with previous methods(e.g. DPMSolver), empirical evaluation have verified the effectiveness of UniPC in generating better results with less sampling steps.

**Strengths:**

-	The proposed method can significantly reduce the number of sampling steps to less than 10, without losing the quality of the results.
-	The proposed method has some generality and can further improve the approximate accuracy of other existing DPM samplers.


**Weaknesses:**

- Experiments. Please conduct experiments on datasets with larger resolution images to measure inference time, quality, and diversity.
- Recent work: Please add more details on the related wok about DPM solver and some related ODE-based sampling solver.


**Questions:**

See weakness.

**Limitations:**

See weakness.

---

> ### Author Rebuttal · Authors · 2023-08-08
>
> We sincerely thank the reviewer for the positive comments on our work! We address the questions and clarify the issues accordingly as described below.
>
> **Q1: About experiments on larger resolution images**
>
> **[Reply]** Thanks for your advice. In our original paper, we have already conducted experiments on $512\times 512$ images (sampling from stable-diffusion). Here we compare our method and the baseline DPM-Solver++ using a larger model _**Stable-Diffusion-XL**_, which can produce $1024\times 1024$ images. We randomly select 200 captions and generate the samples using UniPC and DPM-Solver++ implemented in the diffusers library. Due to the larger resolution, we evaluate both methods with NFE=15, and the results are listed in the following table (the evaluation protocol is the same as the original paper):
>
> |Method|Quality: $\ell_2$-Dist ($\downarrow$)|Diversity: IS($\uparrow$)|Inference Time (s)|
> |------|----------|----------|---------|
> |DPM-Solver++|0.741|8.494|4.33$\pm$0.02|
> |UniPC|0.669|8.909|4.26$\pm$0.01|
>
> Our results show that UniPC achieves better performance in all three metrics, indicating that UniPC can also be a good choice when sampling from a large diffusion model like Stable-Diffusion-XL. We also provide some qualitative results in the attached one-page PDF, where we show that the images generated by UniPC are more realistic and have much fewer artifacts compared with DPM-Solver++.
>
> **Q2: About the recent work**
>
> **[Reply]** Thanks for the suggestion. We will elaborate on the details of the related work and below is another paragraph we plan to add to Section 2.2 (due to a display issue of OpenReview, we cannot type too many formulae here, but we will include more details in the revised paper):
>
> Based on the exponential integrator, [25] proposes to approximate $\hat{\epsilon}_{\theta}$  via Taylor expansion and views DDIM as DPM-Solver-1. [26] considers the data prediction scheme by rewriting (2) and demonstrates its effectiveness in conditional sampling. [40] derives the Taylor expansion formulae with respect to t instead of the half log-SNR. [24] employs pseudo-numerical methods such as the Runge-Kutta method. Although many aforementioned high-order solvers are proposed, existing solvers of diffusion ODEs can be explicitly computed for orders not greater than 3, due to the lack of analytical forms.

---

### Official Review · Reviewer_uABf · 2023-07-10

**Soundness:** 3 good
**Presentation:** 3 good
**Contribution:** 3 good
**Rating:** 6
**Confidence:** 3

**Summary:**

This paper proposes an ODE sampler for diffusion probabilistic models
(DPM), exploiting the structure of exponential integrators.  The paper
claims the proposed sampler can use any existing DPM and achieve high
sampling quality with very few (<10) number of function evaluations
(NFE), and also improve upon related methods when using arbitrary
NFEs.


**Strengths:**

- The proposed method is well justified theoretically and the
  experiments validate the proposed approach.
- The paper clearly states the connections between the proposed method
  and related existing solvers.
- The experimental results include  both pixel-space and
  latent-space diffusion and show consistent improvements when applying
  the method on top of existing solvers.
- The method performance is superior to the baselines in both low and
  middle NFE regimes.


**Weaknesses:**

1) It would be nice if the paper included the statistical significance
   of the FID scores reported in Tables 1-6. In the ~10 NFE regime
   sometimes these are very close to the baseline, while for lower
   quality low NFE one could expect the FID to be not very
   informative. I plan to revisit my decision after considering the
   statitstical significance of these results.
2) The paper repeatedly claims being able to achive higher order p
   than existing solvers. However I wonder how important it is in
   practice to achieve order p>3. Table 4 shows when including higher
   orders the results become poorer.  And also I'd be curious to see
   this study for the higher quality NFE regimes (eg NFE=10).
3) Minor:
   - L. 39 "output of the model output"
   - L. 68-69 extra subscript 0 on lhs?
   - L. 71 "obtained"
   - L. 97 (after period) Can this be rephrased in a less subjective manner?
   - (5) I think $\hat{\epsilon}_\theta^{(k)}$ is not defined. Shouldn't
     the argument in this tem vary with $k$?


**Questions:**

See weaknesses.

**Limitations:**

Yes.

---

> ### Author Rebuttal · Authors · 2023-08-08
>
> We sincerely thank the reviewer for the positive comments on our work! We address the questions and clarify the issues accordingly as described below.
>
> **Q1: About the statistical significance**
>
> **[Reply]** Thanks for the advice. We consider the following hypothesis testing problem:
> $$
> H_0: M_{\rm DPM-Solver++}\le M_{\rm UniPC} \quad \text{versus} \quad H_1: M_{\rm DPM-Solver++}> M_{\rm UniPC}
> $$
> where $M$ is the metric of interest, and a lower value of $M$ implies better performance.  We derive the statistical significance by running our method and the baseline method for independent trials on different datasets and conducting two-sample t-tests for the aforementioned hypothesis testing problem. For CIFAR10, FFHQ, and LSUN, the metric $M$ is FID50K,  and we compute the results of 10 independent runs. For MS-COCO2014, the metric $M$ is the  $\ell_2$ distance between the generated latent codes and the ground truth (obtained by a 999-step DDIM), which is calculated by randomly selecting 200 captions as conditions. Note that for both the two metrics (FID50K and $\ell_2$-Dist), the lower is the better.
>
> The p-values of the two-sample t-tests on different datasets are presented in the following table (where we use NFE=8,10 as examples):
>
> |Dataset \ $p$-value|NFE=8|NFE=10|
> |---------|---------|---------|
> |CIFAR10|$1.09\times 10^{-3}$|$1.46\times 10^{-3}$|
> |FFHQ|$3.01\times 10^{-15}$|$3.66\times 10^{-12}$|
> |LSUN Bedroom|$4.72\times 10^{-4}$| $4.53\times 10^{-5}$|
> |MS-COCO2014|$7.04\times 10^{-5}$|$1.25\times 10^{-5}$|
>
> We can find that all the $p$-values are far smaller than $0.01$, which means that we can reject the null hypothesis of no improvement at the significance of $0.01$. In other words, UniPC performs significantly better than DPM-Solver++.
>
>  **Q2: About the higher order**
>
> **[Reply]** Thanks for pointing out this. We prove theoretically that higher-order solvers enjoy better accuracy in solving a diffusion ODE. In addition, our method offers a unified framework to empirically investigate the performance of higher-order solvers. From our empirical study, the schemes '123432' for NFE 6 and '1223334' for NFE 7 outperform the low-order schemes with $p \leq 3$ and all the existing solvers on CIFAR10. Our results also show that simply increasing the order by introducing more _**previous points**_ might not be beneficial to performance. We speculate that this is because the model output of previous points might be more inaccurate and thus would affect the subsequent sampling steps. On the other hand, our UniC increases the convergence order by re-using the _**current point**_, which can consistently improve the sampling quality over the baseline methods (see Table 2). Here we provide some results of different order schedules when NFE=10 on CIFAR10:
>
> |schedule|1223433321|1233343321|1234544321|
> |------------|-----------------|----------|------------|
> |FID$\downarrow$|4.07 |4.14|4.76|
>
> |schedule|1234554322|1234565432|1234444443|
> |------------|-----------------|----------|------------|
> |FID$\downarrow$|5.41|18.23|6.84|
>
>  **Q3: About the minor issues**
>
> **[Reply]** Thanks for your careful reading. We will modify these as follows:
> - L. 39: "output of the model output" $\rightarrow$ "model output"
> - L. 68-69: We use $q_{t0}$ to represent the transition probability from $x_0$ to $x_t$. We will change it into $q_{t|0}$ for better readability.
> - L. 71: "obtain"  $\rightarrow$ "obtained"
> - L. 97: We will change this sentence into: "Despite the rapid development of fast samplers, the quality in few-step sampling still has room for improvement. "
> - (5): The superscript $k$ denotes the $k$-th derivative of $\hat{\epsilon}_\theta$. We will add the notation in the revised paper.

---

> > ### Comment · Reviewer_uABf · 2023-08-18
> > **Thank you for your rebuttal and the stastistical significance test.**
> >
> > The authors have satisfactorily addressed my concerns in their rebuttal.

---

> > > ### Author Response · Authors · 2023-08-22
> > > **Thanks for the response**
> > >
> > > Thanks a lot for the response. We are glad to hear that our rebuttal has satisfactorily addressed your concerns. Would you please consider raising your score?

---

### Author Rebuttal · Authors · 2023-08-08

We sincerely thank the reviewers for the positive feedback and valuable comments on our work. As suggested by Reviewer vcSP, we further compare the sampling quality of our method UniPC and the baseline DPM-Solver++ using _**Stable-Diffusion-XL**_, a newly released model which can generate $1024\times 1024$ images. We highly encourage the reviewers to have a look at the attached one-page PDF for the qualitative results, where it can be found that our method consistently generates more realistic images with fewer visual flaws compared with DPM-Solver++.

---

> ### Comment · Area_Chair_t2zM · 2023-08-18
>
> Thanks for providing responses. All your comments will be taken into consideration when the committee makes decisions.

---

### Decision · Program_Chairs · 2023-09-21

**Decision:**

Accept (poster)

**Comment:**

This paper aims to improve the sampling speed of the diffusion model with a predictor-corrector framework. It still receives mixed scores (7, 6, 5, 5, 3) after the discussion period. The remaining concerns are about the novelty and experimental performance of this paper. However, this work receives multiple positive comments like " well justified theoretically", " the experiments validate the proposed approach", " reasonably good performance achieved", and " generally written clearly".

Overall, the AC thinks this work can be of large interest to the community working on diffusion model but still hopes the authors emphasize the novelty of this work, as suggested by the reviewers TtDL.